# AGaLiTe: Approximate Gated Linear Transformers for Online Reinforcement Learning

**Subhojeet Pramanik[a,b], Esraa Elelimy[a,b], Marlos C. Machado[a,b,c], Adam White[a,b,c]**

**[a]Department of Computing Science, University of Alberta, Canada**
**[b]Alberta Machine Intelligence Institute (Amii), Canada**
**[c]Canada CIFAR AI Chair**
{spramanik, elelimy, machado, amw8}@ualberta.ca

**Reviewed on OpenReview:** `https://openreview.net/forum?id=lh6vOAHuvo`

## Abstract

In this paper we investigate transformer architectures designed for partially observable online reinforcement learning. The self-attention mechanism in the transformer architecture is capable of capturing long-range dependencies and it is the main reason behind its effectiveness in processing sequential data. Nevertheless, despite their success, transformers have two significant drawbacks that still limit their applicability in online reinforcement learning: (1) in order to remember all past information, the self-attention mechanism requires access to the whole history to be provided as context. (2) The inference cost in transformers is expensive. In this paper, we introduce recurrent alternatives to the transformer self-attention mechanism that offer context-independent inference cost, leverage long-range dependencies effectively, and performs well in online reinforcement learning task. We quantify the impact of the different components of our architecture in a diagnostic environment and assess performance gains in 2D and 3D pixel-based partially-observable environments (e.g. T-Maze, Mystery Path, Craftax, and Memory Maze). Compared with a state-of-the-art architecture, GTrXL, inference in our approach is at least 40% cheaper while reducing memory use more than 50%. Our approach either performs similarly or better than GTrXL, improving more than 37% upon GTrXL performance in harder tasks.

## 1 Introduction

In many real-world settings agents often have limited observability of the environment making decision making extra challenging. For example, an agent designed to drive cars must remember the road signs it saw a few minutes ago to adjust its velocity if there are any major changes to the road. A naive approach would be to store the entire history of camera observations. However, such an approach is not scalable as the history of observations can be longer than the memory available to the agent (McCallum, 1996). Alternatively, the agent can learn a compressed representation of the history of observations, and use it to make decisions. This approach, however, is not feasible in continuing problems where agent's face an unending stream of experience and information critical for decision making occurred in the distant past. Therefore, we need agents that can incrementally update their internal representation of the environment state with computation that does not growth as a function of total experience. In this paper, we investigate incremental state construction in the context of partially observable online reinforcement learning (RL), where agents learn while interacting with the world and thus computation and memory require special consideration.

In RL, incremental state construction is a long-studied problem with many possible solution methods. Recurrent neural network (RNN) architectures provide a framework for learning such representations due to

their ability to automatically learn relationships about the past. RNNs, such as LSTMs (Hochreiter & Schmidhuber, 1997) and GRUs (Gao & Glowacka, 2016), handle sequential data by maintaining a vector of hidden states that capture dependencies between consecutive elements in the sequence. RNNs have been applied to a wide range of partially observable RL environments such as Atari 2600 games (Hausknecht & Stone, 2015) and Starcraft (Vinyals et al., 2017). The inference cost of RNNs—the cost of processing a single element in a sequence of data—is independent of the length of the sequence making them an attractive choice for online RL. Unfortunately, RNNs such as LSTMs are notoriously difficult to train (Bakker, 2001; Khandelwal et al., 2018), and their computations over the input cannot be parallelized.

Transformers (Vaswani et al., 2017) have achieved state-of-the-art performance in many sequential data processing problems, but have seen limited application in online RL (Parisotto et al., 2020). Transformer architectures have been widely used in natural language processing (e.g., Brown et al., 2020; Devlin et al., 2018) and computer vision (e.g., Petit et al., 2021; Zhong et al., 2020). These successes are often attributed to the transformers' self-attention mechanism which can capture long-range dependencies, but they cannot learn relationships that exceed this fixed-length memory. In addition, the inference costs are higher than an RNN: linear in the length of the context window. The linear transformer architecture reduces computational complexity of the self-attention mechanism (Katharopoulos et al., 2020).

In this paper, we introduce two new approaches designed for partially observable RL problems based on the linear transformer's self-attention mechanism. Our new approach to self-attention, based on linear transformers, was designed to achieve the following: (1) a self-attention mechanism that can add and delete previous information, (2) a learned feature map, and (3) a self-attention mechanism that requires computation linear in the size of the embedding dimension. Our first contribution, Gated Linear Transformer (GaLiTe), uses a gated structure that allows it to uncover relationships far in the past. It also uses a different self-attention mechanism that can *learn* a highly parallelizable feature map that is amenable to sequential computation with a context-independent inference cost. Our second contribution, Approximate Gated Linear Transformer (AGaLiTe), introduces an approximate version of GaLiTe's self-attention mechanism, eliminating the need to maintain a matrix as a recurrent state.

We demonstrate the utility of our proposed approaches in several partially observable RL problems. Our experiments show that both GaLiTe and AGaLiTe can match the performance of more computationally expensive transformer architectures in a small diagnostic T-Maze environment. In a pixel-based navigation task, we find that our approach outperforms the state-of-the-art transformer architecture, GTrXL (Parisotto et al., 2020), by more than 37%. Our AGaLiTe-based agent achieves higher rewards than a GTrXL-based agent and higher performance across various in-game skills in Craftax Symbolic (Matthews et al., 2024), a symbolic adaptation of the 2D survival game Crafter (Hafner, 2021). In 3D pixel-based navigation tasks, AGaLiTe's performance is close to GTrXL while reducing the computation and memory by 40% and 50% respectively. Our results in Craftax and 3D navigation provide promising initial evidence of the scalability of our new our approach; a step towards better transformer-based architectures for online, partially observable RL. Code and implementation for this work is publicly available[1].

## 2 Preliminaries

In this section, we provide a brief background on Transformers. We first discuss the canonical transformer architecture and then we discuss the linear transformer approach, which is the basis of our approach.

The transformer architecture was introduced for supervised next token prediction tasks (Vaswani et al., 2017). Our main contribution is a new self-attention mechanism; this section provides the background required to understand the self-attention mechanism in transformers.

Self-attention is mechanically simple. For a given query token $i$ (embedded in $\mathbf{x}_i \doteq \mathbf{X}(\mathbf{i}, \cdot)$), we output an embedded context vector that weights each input token's importance (attention weighted) to the query token. The input to the self-attention layer is a matrix $\mathbf{X} \in \mathbb{R}^{N \times d}$, an embedding of each input token (1 to $N$) into a vector, $\mathbb{R}^d$. The output is a matrix $\mathbf{A} \in \mathbb{R}^{N \times d_h}$, where $d_h$ is the head dimension. Algorithm 1 shows a single self-attention layer with learnable parameters $\mathbf{W}_Q, \mathbf{W}_K, \mathbf{W}_V \in \mathbb{R}^{d \times d_h}$.

---

[1]https://github.com/subho406/agalite

## 2.1 Canonical Transformer Architecture

We can think of the process in two steps. In step one we calculate the attention weights. We compare each token in the context to all other tokens in the context ($\boldsymbol{QK}^T$). The weights are then scaled the size of the embedding dimension and normalized with an element-wise *softmax*. In step two, we compute and return the attention-weighted context vectors, one for each input in $\mathbf{X}$.

---

**Algorithm 1** Canonical Self-Attention

**Input**: $\mathbf{X} \in \mathbb{R}^{N \times d}$
**Parameters**: $\mathbf{W}_Q, \mathbf{W}_K, \mathbf{W}_V \in \mathbb{R}^{d \times d_h}$

1: $\mathbf{Q} \leftarrow \mathbf{XW}_Q$
2: $\mathbf{K} \leftarrow \mathbf{XW}_K$
3: $\mathbf{V} \leftarrow \mathbf{XW}_V$
4: $\mathbf{A} \leftarrow softmax(\frac{\mathbf{QK}^\top}{\sqrt{d}})\mathbf{V}$

**Output**: $\mathbf{A} \in \mathbb{R}^{N \times d_h}$

---

The self-attention mechanism in Algorithm 1 is computationally expensive. We define the inference cost of self-attention as the cost for processing a single element in a sequence. To generate representations for a single element, a query vector is calculated instead of query matrix using a single input (Algorithm 1, step 1). In canonical self-attention mechanism, processing a single element requires having the full sequence as input for generating the value and key matrix (Algorithm 1, step 2 and 3). Thus, the inference cost depends on the input sequence length $N$. For a naive implementation, the inference cost has $\mathcal{O}(Nd^2)$ time and $\mathcal{O}(Nd)$ space complexity; increasing the sequence length linearly increases the computational complexity. A simple mitigation is to limit the size of the input sequence by maintaining a window of the history of input activations in memory (Dai et al., 2019), but doing so limits the past information the self-attention mechanism can recall.

## 2.2 Recurrent Attention with Linear Transformers

The linear transformer architecture (Katharopoulos et al., 2020) introduces a general way of formulating self-attention as a recurrent neural network by replacing the *softmax* with a kernel function, leveraging its equivalence to applying kernel smoothing over inputs (see work by Tsai et al., 2019).

---

**Algorithm 2** Linear Transformer's Self-Attention

**Input**: $\mathbf{x}_t \in \mathbb{R}^d$, $\mathbf{C}_{t-1} \in \mathbb{R}^{d_h \times d_k}$, $\mathbf{s}_{t-1} \in \mathbb{R}^{d_k}$
**Parameters** : $\mathbf{W}_Q, \mathbf{W}_K, \mathbf{W}_V \in \mathbb{R}^{d_h \times d}$
$\mathbf{s}_0 \leftarrow \mathbf{0}, \mathbf{C}_0 \leftarrow \mathbf{0}$.

1: $\mathbf{q}_t \leftarrow \phi(\mathbf{W}_Q\mathbf{x}_t)$
2: $\mathbf{k}_t \leftarrow \phi(\mathbf{W}_K\mathbf{x}_t)$
3: $\mathbf{v}_t \leftarrow \mathbf{W}_V\mathbf{x}_t$
4: $\mathbf{C}_t \leftarrow \mathbf{C}_{t-1} + \mathbf{v}_t \otimes \mathbf{k}_t$
5: $\mathbf{s}_t \leftarrow \mathbf{s}_{t-1} + \mathbf{k}_t$
6: $\mathbf{a}_t \leftarrow (\mathbf{C}_t\mathbf{q}_t)/(\mathbf{s}_t^\top \mathbf{q}_t)$

**Output**: $\mathbf{a}_t \in \mathbb{R}^{d_h}, \mathbf{C}_t \in \mathbb{R}^{d_h \times d_k}, \mathbf{s}_t \in \mathbb{R}^{d_k}$

---

A single time-step of inference of the linear transformer self-attention is described in Algorithm 2. Note that we present the algorithm for processing a single input vector in the case of the linear transformer. This is in contrast to Algorithm 1, which presents the algorithm for processing a sequence. Let $k(\mathbf{a}, \mathbf{b}) = \phi(\mathbf{a})^\top \phi(\mathbf{b})$, where $\phi : \mathbb{R}^{d_h} \rightarrow \mathbb{R}^{d_k}$ is a non-linear feature map, $d_k$ is the output dimension of the feature map $\phi$, and $k : \mathbb{R}^{d_h} \times \mathbb{R}^{d_h} \rightarrow \mathbb{R}^+$. Additionally, let $\otimes$ be defined as the outer product vector operation. At a given timestep $t$, the linear transformer self-attention maintains a matrix, $\mathbf{C}_{t-1} \in \mathbb{R}^{d_h \times d_k}$, and a vector, $\mathbf{s}_t \in \mathbb{R}^{d_k}$, as a recurrent state, which is updated iteratively using the current input vector, $\mathbf{x}_t$. Different from Algorithm 1, Algorithm 2 applies the feature map, $\phi$, to generate the query and key for a given time-step (lines 1 and 2). The linear transformer self-attention stores the outer product of value and key vectors as a recurrent state matrix, $\mathbf{C}_t$ (line 4). Additionally, the sum of the key vectors is stored as a recurrent normalization vector $\mathbf{s}_t$ (line 5). The attention output vector, $\mathbf{a}_t$, is calculated by multiplying the recurrent state with the query vector, and normalizing it using the product of the normalization vector, $\mathbf{s}_t$, and the query vector, $\mathbf{q}_t$ (line 6).

The linear transformer's self-attention has a context-independent inference cost, unlike the canonical self-attention mechanism. In Algorithm 2, processing a single input vector ($\mathbf{x}_t$) has a space and time complexity of $\mathcal{O}(dd_k)$, assuming $d$, the embedding dimension (of the input), is greater than $d_h$, which is the size of the attention-weighted context vector, $\mathbf{a}_t$. Unlike vanilla self-attention, the computational complexity does not depend on the context length, making it more efficient for longer sequences.

## 3 Gated Linear Transformers (GaLiTe)

In this section, we introduce GaLiTe to address two of the limitations of linear transformers. Specifically, (1) the recurrent equations in Algorithm 2 (lines 5 and 6) add positive values to the recurrent state, without any mechanism to delete past information. (2) Performance critically depends on the choice of the kernel feature map $\phi$ (lines 1 and 2); element-wise functions such as the Exponential Linear Unit (ELU) typically perform worse than softmax (Katharopoulos et al., 2020).

GaLiTe mitigates these two issues by introducing a gating mechanism and a parameterized feature map. The gating mechanism controls the flow of information at each index of $\mathbf{C}$ (the location of the recurrent states of the self-attention mechanism), allowing arbitrary context memory (inducing a trade-off with precision). The parameterized feature map is used to calculate the key and query vectors in the self-attention mechanism, eliminating the choice of the kernel feature map $\phi$.

### 3.1 Gating Mechanism to Control the Flow of Information

In the linear transformer self-attention, at a given time-step $t$, Algorithm 2 increments the recurrent state, $\mathbf{C}_{t-1}$, and the normalization vector, $\mathbf{s}_{t-1}$, (lines 4 and 5). Assuming $\mathbf{C}_0$ and $\mathbf{s}_0$ are initialized to zero, recall the update equations for $\mathbf{C}_t$ and $\mathbf{s}_t$ are recursively defined as follows:

$$\mathbf{C}_t \doteq \mathbf{C}_{t-1} + \mathbf{v}_t \otimes \mathbf{k}_t, \qquad (1) \qquad\qquad \mathbf{s}_t \doteq \mathbf{s}_{t-1} + \mathbf{k}_t. \qquad (2)$$

As $\mathbf{k}_t$ is a function of $\phi$ and under the assumption of a positive feature map $\phi$, Equation 2 adds arbitrary positive values to $\mathbf{s}_{t-1}$. Similarly, Equation 1 adds arbitrary positive values to $\mathbf{C}_{t-1}$ if the elements in value vector $\mathbf{v}_t$ are positive. Both Equation 1 and 2 have no way to control the flow of past information and the values in the recurrent state could grow. Instead, we use a normalized exponential average—with element-wise learned decay parameters—which smoothly reduces the impact of past information.

Gating mechanisms can be used to control the flow of information in recurrent updates. We propose a learned outer-product-based gating mechanism that decays every element of $\mathbf{C}_{t-1}$ and $\mathbf{s}_{t-1}$ allowing the network to learn the decay for each element (also known as the memory location). We introduce learnable parameters $\mathbf{W}_\beta \in \mathbb{R}^{d_h \times d}$, $\mathbf{W}_\gamma \in \mathbb{R}^{d_k \times d}$, and gating vectors $\beta_t$, and $\gamma_t$. Let $\sigma_g$ be a sigmoid function defined as $\sigma_g(x) \doteq 1/1+e^{-x}$, we define $\beta_t$ and $\gamma_t$ as follows:

$$\beta_t \doteq \sigma_g(\mathbf{W}_\beta \mathbf{x}_t), \qquad (3) \qquad\qquad \gamma_t \doteq \sigma_g(\mathbf{W}_\gamma \mathbf{x}_t). \qquad (4)$$

Let $\odot$ be the element-wise product, we use the outer product of $\beta_t$ and $\gamma_t$ to control the flow of past information in recurrent states $\mathbf{C}_t$ and $\mathbf{s}_t$, modifying Equations 1 and 2 as follows:

$$\mathbf{C}_t \doteq \big((1 - \beta_t) \otimes (1 - \gamma_t)\big) \odot \mathbf{C}_{t-1} + \big(\beta_t \odot \mathbf{v}_t\big) \otimes \big(\gamma_t \odot \mathbf{k}_t\big), \qquad (5)$$

$$\mathbf{s}_t \doteq (1 - \gamma_t) \odot \mathbf{s}_{t-1} + \gamma_t \odot \mathbf{k}_t. \qquad (6)$$

We use outer products to learn the decay rate for each index of $\mathbf{C}_t$, without requiring individual parameters for each index. The outer product assumes the decay rate at each index is independent from each other.

### 3.2 Learnable Feature Map for Self-Attention

Recall that the self-attention mechanism of the linear transfomer uses a kernel feature map to calculate the key and query vectors:

$$\mathbf{k}_t \doteq \phi(\mathbf{W}_K \mathbf{x}_t), \qquad (7) \qquad\qquad \mathbf{q}_t \doteq \phi(\mathbf{W}_Q \mathbf{x}_t). \qquad (8)$$

We consider a deterministic approach to learn the key and value vectors in the linear transfomer self-attention mechanism. We introduce modifications to $\mathbf{k}_t$, $\mathbf{q}_t$, and gating vectors calculation described in Equations 7, 8, 3, and 4 respectively. We start by introducing a hyperparameter $\eta$ that controls the dimension of the feature maps used to construct $\mathbf{k}_t$ and $\mathbf{q}_t$. Let $\mathbf{W}_{p_1}, \mathbf{W}_{p_2}, \mathbf{W}_{p_3} \in \mathbb{R}^{\eta \times d}$ be learnable parameters. We modify the dimensions of $\mathbf{W}_\gamma$ as $\mathbf{W}_\gamma \in \mathbb{R}^{d_h \times d}$, getting rid of $d_k$, the kernel feature map dimension. Let $f()$ be a function

that flattens a matrix into a vector. We redefine $\mathbf{k}_t$ and $\mathbf{q}_t$ (previously defined in Equations 7 and 8) as follows:

$$\mathbf{k}_t \doteq f(relu(\mathbf{W}_{p_1}\mathbf{x}_t) \otimes relu(\mathbf{W}_K\mathbf{x}_t)) \qquad (9) \qquad \mathbf{q}_t \doteq f(relu(\mathbf{W}_{p_2}\mathbf{x}_t) \otimes relu(\mathbf{W}_Q\mathbf{x}_t)). \qquad (10)$$

We also modify the gating vectors calculation in Equation 4 as follows:

$$\gamma_t \doteq f(\sigma_g(\mathbf{W}_{p_3}\mathbf{x}_t) \otimes \sigma_g(\mathbf{W}_\gamma\mathbf{x}_t)). \tag{11}$$

Using the modified key, query, and gating vectors, the recurrent states $\mathbf{C}_t \in \mathbb{R}^{d_h \times \eta d_h}$ and $\mathbf{s}_t \in \mathbb{R}^{\eta d_h}$ are calculated according to Equations 5 and 6. It is important to note that the feature map dimension, $d_k = \eta d_h$, is now controlled by the hyperparameter $\eta$. Equations 9 and 10 use outer products to learn multiplicative interactions in the key and query vectors. Learning multiplicative interactions in the feature vectors allows learning complex non-linear relationships through training instead of relying on an explicit non-linear element-wise function or on random feature maps. Using outer products to generate an expansive feature map allows us to have a large feature map output dimension. Having a large feature map output dimension is essential as it correlates with the memory capacity (see the work by Schlag et al., 2021).

Finally, we use the relu activation function to ensure the output of the feature map is positive. A positive feature map is a common assumption in the linear transformer literature as it is simple way to ensure that similarity scores produced by the underlying kernel function are positive.

The Gated Linear Transformer (GaLiTe) self-attention incorporates the changes discussed above into the linear transfomer self-attention. The pseudo-code for GaLiTe is available in Appendix A. GaLiTe has similar space and time complexity as the linear transfomer. For processing a single element in a sequence, GaLiTe has a space and time complexity of $\mathcal{O}\left(\eta d^2\right)$ and $\mathcal{O}\left(\eta d^2\right)$, respectively. In comparison, the linear transfomer requires $\mathcal{O}\left(d_k d\right)$ and $\mathcal{O}\left(d_k d\right)$. Notice $d_k$ is defined to be the output dimension of the kernel feature map, which is $\eta d_h$ in GaLiTe. Similar to the linear transfomer, the space and time complexity of GaLiTe is independent of $N$ and only depend on static hyperparameters $d$ and $\eta$.

## 4 Approximate Gated Linear Transformer (AGaLiTe)

Operating on large matrices is expensive. Recall that GaLiTe stores a matrix of dimension $d_h^2 \eta$ as a recurrent hidden state. This becomes more problematic with the use of multiple heads and layers; which are typically required to improve stability during the training (see the work by Michel et al., 2019). For example, previous applications of transformers in RL by Parisotto et al. (2020) use 8 heads and 12 layers; 96 heads in total. Second, the update to $\mathbf{C}_t$ makes use of expensive and memory heavy operations: an outer product, element-wise matrix sum, and multiplication.

Our goal is to approximate the recurrent state update in Equation 5 with an approximation that uses less space than $\mathcal{O}(\eta d^2)$. Recall that Equation 5 replaces $\mathbf{C}_t$ with $\mathbf{C}_{t-1}$ plus a new outer product. To derive an approximation, we want to replace $\mathbf{C}_{t-1}$ with a matrix that has a lower rank. Also, we want to derive an update rule that is an approximation of Equation 5, but instead of updating the full-rank matrix, $\mathbf{C}_{t-1}$, it updates the low-rank approximation.

Our second approach, called Approximate Gated Linear Transformer (AGaLiTe), uses a low-rank approximation to reduce the space complexity of GaLiTe. We replace the previous recurrent state matrix, $\mathbf{C}_{t-1}$, with a set of vectors, reducing the space complexity of GaLiTe by $d$. We introduce an approximation of the Kronecker delta function using a sum of cosine functions and we use this to approximate $\mathbf{C}_{t-1}$.

We introduce an approximation that uses a sum of cosine functions to approximate a sum of outer products. This approximation is deterministic, it does not introduce variance in the approximation, and it keeps incremental updates to the state end-to-end differentiable. Our approach is inspired by the rank-1 approximation introduced by Ollivier et al. (2015), but instead of using random numbers to approximate a Kronecker delta function, we use a trigonometric identity that relates a Kronecker delta function to an integral over cosines. Recall that the Kronecker delta function is defined for integers $m$ and $n$ such that $\delta_{mn} = 1$ if $m = n$, and

$\delta_{mn} = 0$ if $m \neq n$. We present an approximation $\hat{\delta}_{mn}$ of $\delta_{mn}$ such that $\hat{\delta}_{mn}$ is defined as follows:

$$\hat{\delta}_{mn} \doteq \frac{2}{r} \sum_{i=0}^{r} \left( \cos\left(\frac{2\pi i}{r} m\right) \cos\left(\frac{2\pi i}{r} n\right) \right). \tag{12}$$

It can further be shown that $\lim_{r \to \infty} \hat{\delta}_{mn} = \delta_{mn}$. The derivation for this result is presented in Appendix B.1. We use the approximation of the Kronecker delta function in Equation 12 to approximate the recurrent state update in Equation 5. Briefly, the approximation introduces the approximate Kronecker delta function to approximate $\mathbf{C}_t$ as a sum of $r$ outer-products, where each of the vectors in the outer-product is defined recursively and updated using the value and key at the current timestep. For a given $r$, we maintain recurrent states $\tilde{\mathbf{v}}_{t-1}^k$ and $\tilde{\mathbf{k}}_{t-1}^k$ for $k = 0, 1, \ldots, r$. For $\omega_k \doteq \frac{2\pi k}{r}$, and assuming $\tilde{\mathbf{v}}_0^i$ and $\tilde{\mathbf{k}}_0^i$ are initialized as zeros, we directly calculate the attention output, $\mathbf{a}_t$, in replacement of $\mathbf{C}_t$, considering the recurrent updates to $\tilde{\mathbf{v}}_t^i$ and $\tilde{\mathbf{k}}_t^i$:

$$\tilde{\mathbf{v}}_t^k \doteq \cos(\omega_k t)\beta_t \odot \mathbf{v}_t + (1 - \beta_t) \odot \tilde{\mathbf{v}}_{t-1}^k, \quad (13) \qquad \tilde{\mathbf{k}}_t^k \doteq \cos(\omega_k t)\gamma_t \odot \mathbf{k}_t + (1 - \gamma_t) \odot \tilde{\mathbf{k}}_{t-1}^k, \quad (14)$$

$$\mathbf{a}_t \doteq \frac{\sum_{k=0}^{r} \tilde{\mathbf{v}}_t^k \left( \left(\tilde{\mathbf{k}}_t^k\right)^\top \mathbf{q}_t \right)}{2r(\mathbf{s}_t^\top \mathbf{q}_t)}. \tag{15}$$

Due to space constraints, the rationale behind these approximations is presented in Appendix B.1.1.

The pseudocode for AGaLiTe can be found in Appendix C. Unlike Equation 5, Equations 13 and 14 define a recurrence over vectors instead of matrices. If $r \ll d$, then the recurrence is more efficient in space than the recurrence in Equation 5. In Appendix E, we provide an empirical evaluation of the impact of different values of $r$ in the quality of the approximation, showing that, in practice, it seems small values of $r$ do not compromise the quality of the approximation or the overall performance. The computational complexity of AGaLiTe is $\mathcal{O}(r\eta d)$ and $\mathcal{O}\left(d^2 + r\eta d\right)$ in space and time. With AGaLiTe, we have significantly improved the complexity of the self-attention mechanism and these differences manifest in experiments as we show next. We compare the computational complexities of our

Table 1: Space and time complexity of AGaLiTe, GaLiTe, linear transformer, and GTrXL for processing a single element in a streaming sequence ($M$: memory size in GTrXL, $d$: representation dimension, $d_k$ feature map dimension in the linear transformer, $\eta$: feature map hyperparameter in GaLiTe and AGaLiTe, $r$: approximation parameter in AGaLiTe).

|  | Space | Time |
|---|---|---|
| GTrXL | $\mathcal{O}(Md)$ | $\mathcal{O}(M\,d^2)$ |
| Linear Transformer | $\mathcal{O}(d_k d)$ | $\mathcal{O}(d_k d)$ |
| GaLiTe | $\mathcal{O}\left(\eta d^2\right)$ | $\mathcal{O}\left(\eta d^2\right)$ |
| AGaLiTe | $\mathcal{O}(r\eta d)$ | $\mathcal{O}\left(d^2 + r\eta d\right)$ |

proposed approaches and GTrXL (Parisotto et al., 2020) in Table 1. We provide empirical latency measurements of forward pass using the AGaLiTe architecture and compare it to GTrXL in Appendix K. We also discuss the parallelization of GaLiTe and AGaLiTe over a sequence of data in Appendix F.

## 5 Empirical Evaluation

This section investigates our proposed approaches in several partially observable reinforcement learning (RL) control problems. The memory requirements vary across the environments we consider. In T-Maze (Bakker, 2001), the agent must remember a single cue signal. In CartPole, the agent must estimate the hidden state by integrating information over time. In Mystery Path (Pleines et al., 2023), the agent must remember multiple locations in a grid environment. In Craftax (Matthews et al., 2024), a 2D survival game, the agent faces with partial observability as it can only observe a limited portion of a large 2D map. In Memory Maze environment (Pašukonis et al., 2023), the agent must retain the layout of a 3D maze in addition to several locations across the maze.

Additionally, we also provide results in Long Range Arena (Tay et al., 2021) in Appendix D, a classical benchmark used to evaluate the ability to learn long-range dependencies in a supervised learning scenario. We

evaluate in two of the tasks from the benchmark: ListOps and IMDB. We found that AGaLiTe outperforms transformers and linear transformers across both of these tasks, despite using a smaller number of learnable parameters.

**Diagnostic MDP.** The T-Maze environment is used to evaluate an agent's ability to learn long context dependencies in a reinforcement learning scenario (Bakker, 2001). In this environment, the agent must remember a cue shown only at the beginning of an episode in order to decide which way to turn at the end of a hallway (inset plot in Figure 1). The cue is only included in the observation on the first timestep. The difficulty of this environment can be increased by increasing the corridor length. The agent's actions are NSEW, and the observation is a binary encoding of the current cell (gray code), the cue (on the first step), and several random distractor bits. The full details are provided in Appendix G.1.

We trained seven agents for five million steps in the T-Maze environment, for corridor lengths 120–200. The network architecture for each agent has a shared representation learning layer, either an RNN or a transformer, which is then followed by separate actor and critic heads. Two of these agents were trained using an RNN as the shared representation layer, namely LSTM (Hochreiter & Schmidhuber, 1997) and GRU (Cho et al., 2014). The other two agents used a transformer, particularly the GTrXL architecture (Parisotto et al., 2020), and the linear transformer architecture (Katharopoulos et al., 2020). In GTrXL, the memory size hyperparameter, defined as the amount of stored history, controls the context length. We train two GTrXL agents, GTrXL-128 and GTrXL-256, corresponding to memory sizes 128 and 256. Note that for the corridor lengths considered, GTrXL-256 has the entire episode provided as input. We also evaluate GaLiTe ($\eta = 4$) and AGaLiTe ($\eta = 4, r = 1$); we do so by replacing the XL-attention (Dai et al., 2019)

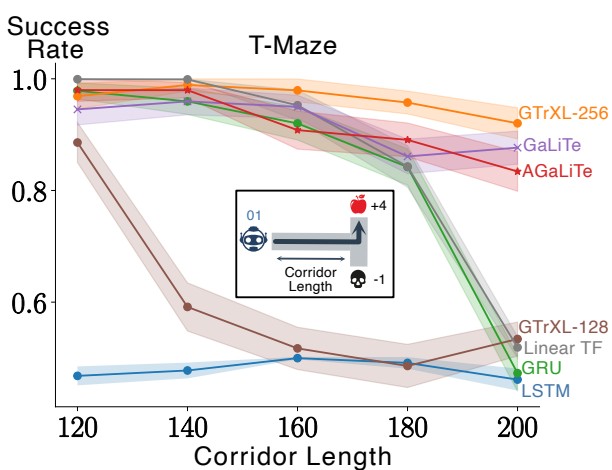

Figure 1: Success rate in the last 100K timesteps averaged over 50 runs in T-Maze (shown inset). The shaded regions represents the standard error.

of GTrXL with one of the two approaches, while preserving the order of the layers and the gating of GTrXL. This allows us to evaluate exactly the impact of the newly introduced self-attention mechanisms without other confounders. The base RL algorithm for all agents use Advantage Actor-Critic (A2C) (Wu et al., 2017). Architecture-specific hyperparameters and tuning strategies are described in Appendix G.1.

Figure 1 summarizes the main results. We report the success rate, the percentage of correct decisions, averaged over the last 100K timesteps of the experiment. An agent that chooses randomly at the intersection would achieve a success rate of 0.5. In this experiment, GTrXL is sensitive to the amount of history provided as input; GTrXL-128 (brown) fails for corridor lengths greater than 120, whereas GTrXL-256 (orange) works well across all corridor lengths. GaLiTe (purple) and AGaLiTe (red) match the performance of GTrXL-256 despite not having access to the entire episode as input. Note that AGaLiTe performs close to GaLiTe even with $r = 1$ (the approximation parameter). GRU (green) and linear transformer (grey) outperform LSTM (blue), but their performance drop in the longest corridor lengths. AGaLiTe is more computationally efficient than GTrXL-256 in T-Maze. For a single attention head, AGaLiTe uses roughly 125.1 times fewer operations than GTrXL-256, and 36.57 times less space. Also, AGaLiTe uses roughly 62.67 times fewer operations and 18.28 times less space than GTrXL-128.

**Partially Observable Classic Control.** Inspired by previous work (Morad et al., 2022; Duan et al., 2016), we explored two variants of partially observable CartPole (Barto et al., 1983). In the first variant, we mask out the velocity information from the observation vector and only allow positional information. This modification makes the problem difficult as the agent now needs to estimate these velocities itself. The second modification introduced an additional challenge by adding noise to the positional information communicated

to the agent. We sampled the noise from a normal distribution with zero mean and 0.1 standard deviation. This problem is qualitatively different from the T-Maze because of the different requirements imposed by the environment. In CartPole, the agents must integrate information over time to construct a reasonable estimate of the underlying state of the MDP, whereas in T-Maze the agent must learn the cue was important and remember it for a long period of time.

We used both GRU and GTrXL as baselines for this problem. GRU-based approaches perform best on these partially observable classical control tasks, even compared to transformers (Morad et al., 2022). We used PPO (Schulman et al., 2017) and trained all agents for 5M steps on the two variants of Cartpole. We also performed an extensive sweep of the hyperparameters of PPO and GRU, which is described in Appendix G.2.

Figure 2 summarizes the results of our experiment in the Noisy stateless CartPole, the second variant from above. The AGaLiTe agent learns faster and finds a better-balancing policy than the GRU and GTrXL agents. The results for the other variant of partially observable CartPole (without noise) is qualitatively similar and can be found in Appendix G.2.

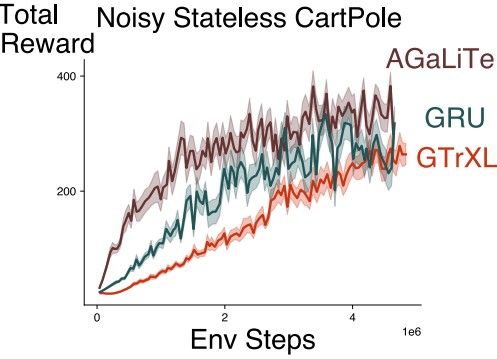

Figure 2: The total reward is binned over 10 timesteps and averaged over 30 different seeds ± standard error.

**Mystery Path.** In Mystery Path (Pleines et al., 2023), the agent is required to remember multiple cue signals for long periods of time in a 2D pixel-based environment. In this environment, the agent's goal is to reach a target position by traversing through a random invisible path. Episodes have fixed length and the agent is reset back to the start location (along with a feedback observation) upon deviating from the path. We consider two configurations of this environment: MPGrid and MP; MP is more difficult. In MP, there are six actions and smoother motion dynamics compared to MPGrid, with grid-like movements and four actions. MPGrid has a maximum episode length of 128, while MP's is 512. Appendix G.3 describes the environment and the configurations considered.

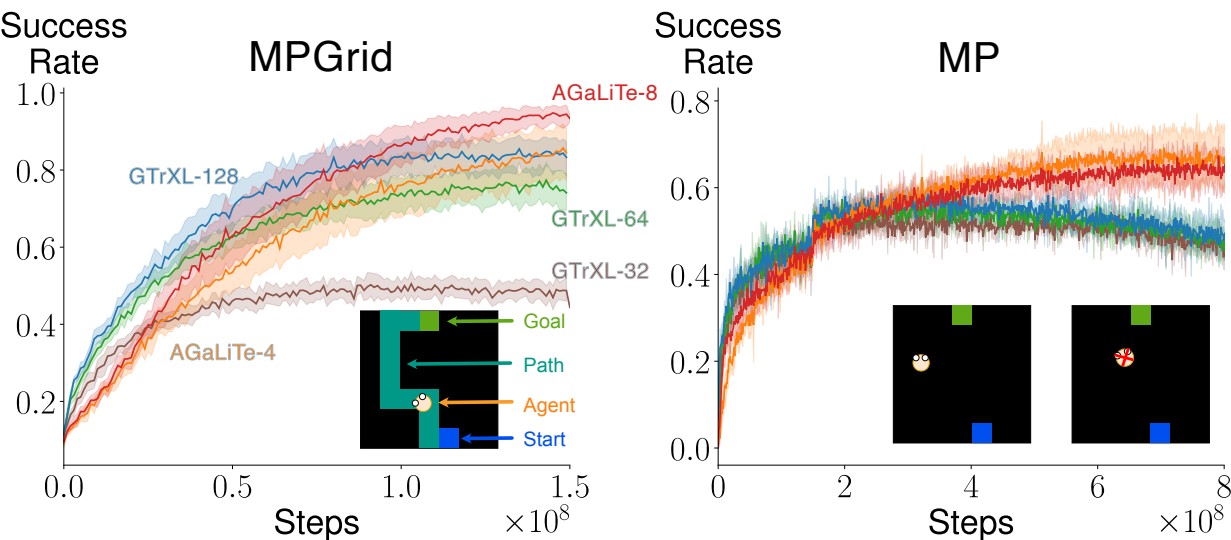

Figure 3: Left: Learning curves in MPGrid (averaged over 15 seeds ± 95% bootstrapped CI) along with an inset figure showing a possible ground truth maze layout. Right: Learning curves in MP (averaged over 5 seeds ±95% bootstrapped CI) along with inset figure depicting the agent's observation. The agent does not observe the path to the goal (left); a red cross is shown as feedback if the agent deviates off from the path, with the agent being reset to the start tile (right).

We trained three GTrXL agents with memory sizes $\in \{32, 64, 128\}$, and two AGaLiTe agents with feature map dimension $\eta \in \{4, 8\}$, and $r = 1$. The architecture sizes for GTrXL and AGaLiTe were chosen to be similar to the ones used in the T-Maze experiments. PPO was the base RL agent used. We used a standard agent network architecture (e.g., Mnih et al., 2016; Schulman et al., 2017) for all agents. Details on hyperparameters sweeps can be found in Appendix G.3.

Figure 3 summarizes the main results. Again we report success rate, the percentage of episodes the agent reaches the goal before an episode timeout, calculated over a window of one million steps. Across both configurations, MPGrid and MP, we observe that AGaLiTe matches the performance of GTrXL-128 when $\eta = 4$ and surpasses GTrXL-128 in mean performance when $\eta = 8$. Also, similar to T-Maze, we observe that reducing the memory size of GTrXL drastically impacts its performance.

We observe again that AGaLiTe is more computationally efficient than GTrXL. For a single attention head, AGaLiTe-8 uses roughly 55.75 times fewer operations than GTrXL-128, and it uses 9.84 times less space. In other words, we observe performance at least as good as GTrXL, in both variants of pixel-based control, at a fraction of the cost.

**Craftax.** Crafter (Hafner, 2021) is a 2D open-world survival game where an agent needs to forage for food and water, find shelter to sleep, defend against monsters, collect materials, and build tools. This environment is designed to evaluate an agent's ability to perform a wide range of tasks purely from a scalar reward signal. The environment is partially observable as the agent can only observe a portion of the large randomly generated map that it navigates. Our hypothesis is that an agent with better memory capabilities and longer context should achieve higher performance through navigating and utilizing the cues in the environment effectively. We consider the symbolic variant of the environment, Craftax symbolic, detailed in Matthews et al. (2024). The symbolic variant simplifies the observations of original pixel-based crafter by encoding each pixel as a one-hot vector and an intensity value.

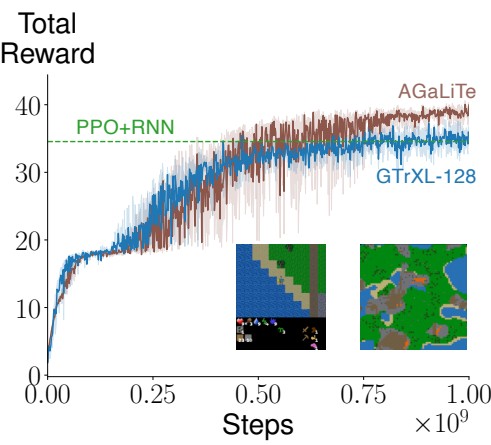

Figure 4: Learning curves of GTrXL-128 and AGaLiTe in the Craftax symbolic environment (averaged over 15 seeds $\pm$ 95% bootstrapped CI). The inset plot shows the result of rendering a sample observation (left) and the full map (right).

We trained a GTrXL agent with memory size 128 and an AGaLiTe agent with feature map dimension $\eta = 8$, and $r = 1$ for 1B steps. The architecture sizes for GTrXL and AGaLiTe were chosen similar to the ones used in the T-Maze and Mystery Path experiments. PPO was the base RL agent used. Details on hyperparameters sweeps can be found in Appendix G.5.

Figure 4 shows the total episodic reward achieved by the AGaLiTe-based agent compared with a GTrXL-based agent. We observe that the AGaLiTe achieves a higher reward than both GTrXL and previously reported PPO+RNN baseline (Matthews et al., 2024). In Appendix J we compare the performance across the various game-related achievements present in Crafter. We find that AGaLiTe achieves higher scores in several of these achievements.

We also considered a 3D navigation environment called Memory Maze (Pašukonis et al., 2023) that has a fixed horizon and that requires the agent to remember multiple cue signals for long periods of time. At the beginning of each episode, a new maze is generated randomly and several objects of different colors are distributed across the maze. The agent perceives a $64 \times 64$ RGB image with a colored border indicating the color of the current object of interest. Once the agent touches the object, it gets a $+1$ reward and the borders' colors change. The agent's goal is to maximize rewards within the fixed time budget. Thus, the agent must remember the objects' locations to travel through the maze as quickly as possible. Figure 5 (inset) provides an illustration of the Memory Maze environment. In the main paper, we report results on the largest maze size, $15 \times 15$, with an episode duration of 4,000 steps. Results for other maze sizes can be found in Appendix H.

**Memory Maze.** We trained a GTrXL agent and an AGaLiTe agent, each with 22M learnable parameters, for 100M steps using the Async-PPO algorithm (Petrenko et al., 2020). The GTrXL agent had a memory size of 256, and the AGaLiTe agent had a feature map $\eta = 4$ and an approximation hyperparameter $r = 7$. We based our architectures for both the policy and the critic on the work by Petrenko et al. (2020). In this work, a ResNet (He et al., 2016) is used to extract the features from the input image, then a sequence of features are fed into an RNN or a transformer. We detail the hyperparameters used, the architecture sizes, and the tuning strategy in Appendix G.4.

Figure 5 shows the total episodic reward achieved by our AGaLiTe-based agent compared with a GTrXL-based agent. The asymptotic performance of all the three agents is similar, but the GTrXL-based agent exhibits faster learning early on. Additionally, we explore the impact of smaller GTrXL memory sizes in Appendix I. We find that reducing the memory size of GTrXL did not affect the performance in this environment, suggesting that the agents might not be utilizing their memory capabilities effectively.

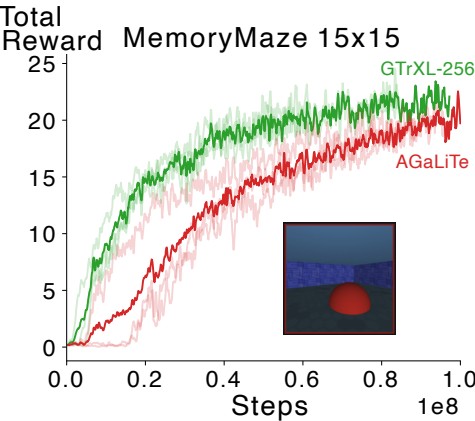

Figure 5: Learning curves in MemoryMaze . The bold lines report total episodic reward averaged over three seeds; the other lines are individual seeds. The inset plot shows a sample observation.

Finally, we looked at the agents' utilization of the computational resources. For a single attention head, AGaLiTe uses roughly 125 times fewer operations than GTrXL-256 and it uses 46 times less space. Additionally, we measured the frames per second (FPS) and the memory usage from 12 AGaLiTe and GTrXL agents. Overall, AGaLiTe achieves $535.63 \pm 0.52$ FPS while GTrXL achieves $373.63 \pm 0.49$ FPS, corresponding to a 43.36% improvement. Further, AGaLiTe uses 52.37% less memory than the GTrXL agent. The number of operations and space used are asymptotic information that highlight the benefits one can expect when using even bigger neural network architectures, such as those now common in industry. The FPS rate demonstrates the performance gain when AGaLiTe is instantiated in a particular network architecture.

## 6 Ablation Study

In this section, we present ablations for each of the three modifications proposed to the linear transformer architecture in AGaLiTe, highlighting the importance of each modification. We present the results for these ablations in Figure 6.

Our first ablation evaluates the impact of proposed gating mechanisms in AGaLiTe. We conduct this ablation in the MPGrid environment. This environment requires an agent to selectively filter and remember multiple pieces of information throughout an episode. Our proposed gating mechanism significantly outperforms an AGaLiTe without the gating mechanism in Figure 6a.

The other two ablations compare the proposed feature map and approximation approach to other approaches in the literature. We conducted these ablations in the T-Maze environment with the corridor length set to 200. We use the same hyperparameter tuning strategy as described in Section G.1, but focusing on the best hyperparameter on corridor length 200. We report the success rate while training an agent for 5M steps over 50 seeds.

To evaluate the impact of different feature maps $\phi$ in AGaLiTe, we consider two alternatives, proposed in the existing literature. The first uses an element-wise feature map $ELU + 1$ (Clevert et al., 2016), which was used originally in the linear transformer architecture. The second is the deterministic parameter free projection (DPFP) introduced by Schlag et al. (2021), which was shown to outperform exisiting feature map approaches in language modelling tasks. We present these results in Figure 6b. We observed that our proposed feature map outperform both of these methods.

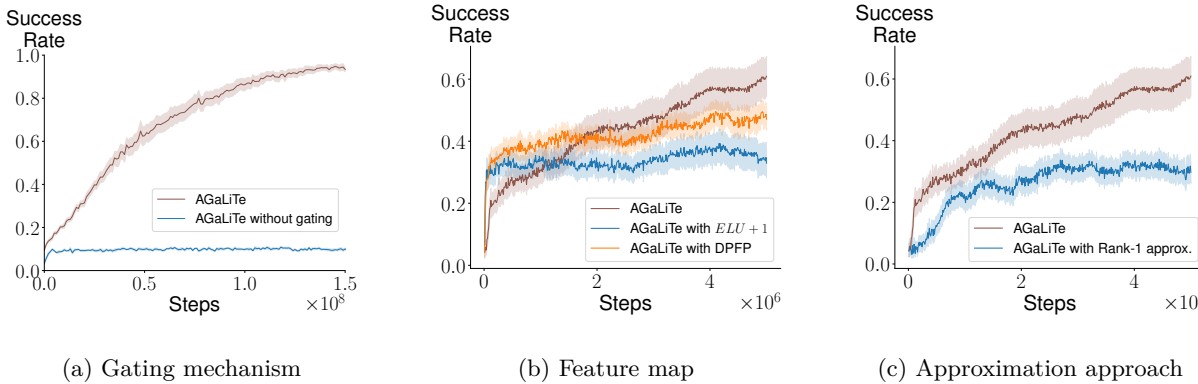

(a) Gating mechanism      (b) Feature map      (c) Approximation approach

Figure 6: Ablation of various components of the AGaLiTe architecture

Finally, we compare AGaLiTe's approximation to an alternative incremental low-rank approximation method. We consider the rank-1 trick introduced by Ollivier et al. (2015). The rank-1 trick approximates a Kronecker delta function using random signs drawn from a uniform distribution. Similar to our proposed approximation approach, the rank-1 trick could be applied to derive incremental updates to a low-rank decomposition of a matrix. We derived an approximation using the rank-1 trick and compared it to our proposed approximation (with $r = 1$) in Figure 6c. We observe that our proposed approximation approach outperforms the rank-1 trick.

# 7 Related Work

Similar to our approach, several previous works have explored extensions to the linear transformer architecture, addressing its limitations. The DeltaNet architecture (Schlag et al., 2021) implements a scalar gating mechanism to accumulate the value vectors over time, and then uses an error-correcting delta rule to update the recurrent states. In contrast, our proposed approach utilizes an element-wise gating mechanism applied directly over the recurrent states. DeltaNet also introduces a deterministic feature map called DPFP we have found to perform worse compared to our proposed feature map in Figure 6. RecurrentDeltaNet architecture (Irie et al., 2021) proposed improvements to the DeltaNet architecture by introducing additional recurrence and non-linearity to the updates of the key, value, and query vectors. However, the non-linear update rule in RecurrentDeltaNet limits its parallelizability over an input sequence. In contrast, our proposed approach uses linear update rules, and could be parallelized over an input sequence (Appendix F). In a concurrent and independent work, Aksenov et al. (2024) explored a learnable kernel function inspired from Taylor expansion of exponential functions, demonstrating impressive in-context learning performance. Notably, none of these work improve upon the computational complexity or introduce approximations to recurrence mechanism of the linear transformer.

Gating mechanisms such as the one we used in GaLiTe and AGaLiTe are commonly used in RNNs to control the flow of information and mitigate the impact of vanishing gradients (Hochreiter & Schmidhuber, 1997). Often, scalar gating mechanisms have been applied, such as in the linear transformer (Peng et al., 2021). However, using a single learned coefficient could be sub-optimal as it does not allow for a more fine-grained control of the flow of past information from each index location in a recurrent state.

The choice of the feature map $\phi$ can have a significant impact on the overall performance (Schlag et al., 2021). For example, a non-expansive map based on *ELU+1* can be used (Katharopoulos et al., 2020), however, element-wise activation functions are limited in their ability to learn complex non-linear relationships and using them as a feature map limits the memory capacity of the architecture (Schlag et al., 2021). Alternatively, random feature maps can be used to approximate a softmax function (Peng et al., 2021; Choromanski et al., 2021). Although randomized feature maps are equivalent to softmax function in expectation, they introduce additional variance. Our model is deterministic.

In the context of AGaLiTe, there are other incremental approaches to approximating large matrices. Incremental singular value decomposition (SVD) (Brand, 2002; 2006) provides a way to perform additive modifications to a low-rank singular value decomposition of a matrix. Previous applications of incremental SVD in RL, however, suggest that sensitivity to the rank parameter is a significant issue (Pan et al., 2017). The rank-1 approximation introduced by Ollivier et al. (2015) uses random numbers to approximate a Kronecker delta function producing an unbiased approximation of a matrix represented as a sum of outer products. The use of random numbers, however, introduces variance in the approximation (Cooijmans & Martens, 2019); our results in the T-Maze suggest the proposed approximation leads to better results than the rank-1 approximation.

Various approaches have been proposed that increase the context length of the transformers or reduce the computational complexity of self-attention. The LongFormer architecture (Beltagy et al., 2020) selectively calculates a sparse attention matrix over a large context and the Transformer-XL architecture (Dai et al., 2019) caches previous activations to attend over a larger context. Alternatively, the RMT architecture (Bulatov et al., 2022) introduced segment-level recurrence to pass global information over a larger context. Other approaches reduce the computational complexity of self-attention by proposing approximations (Kitaev et al., 2020; Choromanski et al., 2021; Wang et al., 2020).

Other methods such as RWKV (Peng et al., 2023), and state-space models such as LRU (Orvieto et al., 2023), S4 (Gu et al., 2021), and S5 Smith et al. (2023) offer context-independent inference cost while leveraging parallelization over a sequence. The S4 architecture was recently applied to offline in-context RL (Lu et al., 2024) and model-based RL (Samsami et al., 2024), demonstrating superiority over RNNs. However, none of these approaches have yet been explored in the model-free RL setting, which has been the main focus of this paper.

Several works have explored using transformers in RL. Parisotto & Salakhutdinov (2021) used transformers to learn policies in an asynchronous setting relying on policy distillation to make interaction with the environment feasible. Others have explored transformers in model-based, fully-observable RL, such as the TransDreamer architecture which replaces the GRU used inside Dreamer V2 (Hafner et al., 2020) with a transformer (Chen et al., 2022).

While focus of this paper has been towards online RL settings, several previous works have applied transformers to offline RL and in-context RL. Chen et al. (2021) demonstrated that transformers could learn single-task policies from offline RL data through imitation learning. Extensions to Chen et al. (2021) demonstated that transformers could also derive multi-task policies from offline RL data in both same-domain (Lee et al., 2022) and cross-domain environments (Reed et al., 2022). Laskin et al. (2023) and Lin et al. (2024) demonstrated that transformers could be trained on offline datasets to learn RL policies in-context on novel tasks. Offline training of transformers can also be combined with online RL and has demonstrated application in discovering high value molecules (Ghugare et al., 2024). We believe that our proposed approach could be also be useful across offline and in-context RL settings by offering a larger context with reduced computational requirements.

## 8 Conclusion and Future Work

Transformers have revolutionized many branches of AI research, but their computational requirements make extension to other domains such as online RL difficult. In this paper, we have introduced two recurrent alternatives of the self-attention mechanism in transformers, called Gated Linear Transformer (GaLiTe) and Approximate Gated Linear Transformer (AGaLiTe). We demonstrate the efficacy of both approaches in a several partially observable reinforcement learning tasks (e.g., T-Maze, Mystery Path, Craftax, Memory Maze). When compared to a state-of-the-art architecture GTrXL, the inference cost of our approach is more than 40% cheaper while reducing memory use more than 50%.

Future work could explore algorithmic improvements to AGaLiTe such as using updates based on efficient real-time recurrent learning (Zucchet et al., 2023; Williams & Zipser, 1989). Furthermore, the application of AGaLiTe to model-based RL algorithms such as the Dreamer V3 (Hafner et al., 2023) could be exciting. Finally, Morad et al. (2024) leverages the properties of linear transformer approaches to propose a batching

method that improves sample efficiency, increases the return, and simplifies the implementation of recurrent loss functions in RL. Application of some of these ideas to AGaLiTe would be exciting. In addition, previous work has found RNN-based approaches are best in some tasks and transformers better in others. There is much to be understood empirically in partially observable RL.

## Acknowledgements

We would like to thank Martha White, Dale Schuurmans, and Michael Bowling for providing valuable feedback and for their helpful discussions. We would like to thank Martha White for also providing access to additional computational resources. We would like to thank Vincent Liu for providing feedback on the derivations presented in this paper. The research is supported in part by the Natural Sciences and Engineering Research Council of Canada (NSERC), the Canada CIFAR AI Chair Program, the University of Alberta, Google Cloud Incubator, TPU Research Cloud Program, and the Digital Research Alliance of Canada.

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

## A  Gated Linear Transformers (GaLiTe)

Algorithm 3 formalizes the self-attention mechanism introduced in GaLiTe. The algorithm introduces a hyperparameter, $\eta$, and a few learnable parameters, $\mathbf{W}_\beta, \mathbf{W}_\gamma \in \mathbb{R}^{d \times d_h}$, and $\mathbf{W}_{p_1}, \mathbf{W}_{p_2}, \mathbf{W}_{p_3} \in \mathbb{R}^{d \times \eta}$. The hyperparameter $\eta$ controls the size of the recurrent states, $\mathbf{C}_t$ and $\mathbf{s}_t$, and of the key and the query vectors.

---

**Algorithm 3** Gated Linear Transformer (GaLiTe) Self-Attention

---

**Input**: $\mathbf{x}_t \in \mathbb{R}^d$, $\mathbf{C}_{t-1} \in \mathbb{R}^{d_h \times \eta d_h}$, $\mathbf{s}_{t-1} \in \mathbb{R}^{\eta d_h}$
**Hyperparameters:** $\eta$
**Parameters**: $\mathbf{W}_K, \mathbf{W}_Q, \mathbf{W}_V, \mathbf{W}_\beta, \mathbf{W}_\gamma \in \mathbb{R}^{d_h \times d}$ and $\mathbf{W}_{p_1}, \mathbf{W}_{p_2}, \mathbf{W}_{p_3} \in \mathbb{R}^{\eta \times d}$

 1: **if** $t = 0$ **then**
 2:     $\mathbf{s}_0 \leftarrow \mathbf{0}, \mathbf{C}_0 \leftarrow \mathbf{0}$.
 3: **end if**

{Calculate Key}

 4: $\mathbf{k}_t \leftarrow f(relu(\mathbf{W}_{p_1}\mathbf{x}_t) \otimes relu(\mathbf{W}_K\mathbf{x}_t))$

{Calculate Query}

 5: $\mathbf{q}_t \leftarrow f(relu(\mathbf{W}_{p_2}\mathbf{x}_t) \otimes relu(\mathbf{W}_Q\mathbf{x}_t))$

{Calculate Value}

 6: $\mathbf{v}_t \leftarrow \mathbf{W}_V\mathbf{x}_t$

{Generate Gating Vectors}

 7: $\beta_t \leftarrow \sigma_g(\mathbf{W}_\beta\mathbf{x}_t)$
 8: $\gamma_t \leftarrow f(\sigma_g(\mathbf{W}_{p_3}\mathbf{x}_t) \otimes \sigma_g(\mathbf{W}_\gamma\mathbf{x}_t))$

{Update Memory}

 9: $\mathbf{C}_t \leftarrow \big((1-\beta_t) \otimes (1-\gamma_t)\big)\odot\mathbf{C}_{t-1} + \big(\beta_t\odot\mathbf{v}_t\big) \otimes \big(\gamma_t\odot\mathbf{k}_t\big)$
10: $\mathbf{s}_t \leftarrow (1-\gamma_t)\odot\mathbf{s}_{t-1} + \gamma_t\odot\mathbf{k}_t$

{Calculate Attention Vector}

11: $\mathbf{a}_t \leftarrow (\mathbf{C}_t\mathbf{q}_t)/(\mathbf{s}_t\mathbf{q}_t)$

**Output**: $\mathbf{a}_t \in \mathbb{R}^{d_h}$, $\mathbf{C}_t \in \mathbb{R}^{d_h \times \eta d_h}$, $\mathbf{s}_t \in \mathbb{R}^{\eta d_h}$

---

## B  Derivation of AGaLiTe

In this section, we walk through the derivations to approximate the GaLiTe self-attention mechanism. We first start with deriving an approximation for the Kronecker Delta Function and then use these approximation results to derive the AGaLiTe's self-attention mechanism.

### B.1  Approximation of Kronecker Delta Function

In this section we derive an approximation of the Kronecker delta function. The Kronecker delta function, $\delta_{mn}$ is defined for integers $m$ and $n$ as 1 if $m = n$ and 0 if $m \neq n$.

We use a trigonometric identity that is used in computing Fourier series by relating the Kronecker delta function to an integral of a product of two cosine functions (Weisstein). It is given by:

$$\delta_{mn} = \frac{1}{\pi} \int_0^{2\pi} \cos(mx)\ \cos(nx)\ dx. \tag{16}$$

We use the Trapezoidal rule to approximate the integral in Equation 16. The trapezoidal rule is a numerical integration method that approximates the integral of a function by dividing the interval into sub-intervals and approximating the function in each sub-interval with a straight line connecting the endpoints. For a function $f(x)$ that is integrable on the interval $[a, b]$, it is given by:

$$\int_a^b f(x) \, dx \approx \sum_{k=1}^{r} \frac{f(x_{k-1}) + f(x_k)}{2} \Delta x, \tag{17}$$

where $\Delta x = \dfrac{b - a}{r}$, $x_k = a + k\Delta x$, and $r$ is the number of sub-intervals used for the integral and it controls the degree of approximation. As $r \to \infty$, the approximation becomes exact. Let $\tilde{\delta}_{mn}$ be the Trapezoidal approximation of the integral defined in Equation 16. We then write $\tilde{\delta}_{mn}$ as:

$$\tilde{\delta}_{mn} = \frac{1}{r} \sum_{i=0}^{r-1} \cos\left(\frac{2\pi i}{r} m\right) \cos\left(\frac{2\pi i}{r} n\right) + \frac{1}{r} \sum_{i=1}^{r} \cos\left(\frac{2\pi i}{r} m\right) \cos\left(\frac{2\pi i}{r} n\right) \tag{18}$$

Further, in the limit we have: $\lim_{r\to\infty} \tilde{\delta}_{mn} = \delta_{mn}$.

Next, we will simplify the above equation to combine the two summations above into a single one:

$$\tilde{\delta}_{mn} = \frac{1}{r} \sum_{i=0}^{r-1} \cos\left(\frac{2\pi i}{r} m\right) \cos\left(\frac{2\pi i}{r} n\right) + \frac{1}{r} \sum_{i=1}^{r} \cos\left(\frac{2\pi i}{r} m\right) \cos\left(\frac{2\pi i}{r} n\right)$$

Adding and subtracting $\dfrac{1}{r}(\cos(0)\cos(0) + \cos(2\pi m)\cos(2\pi n))$:

$$\begin{aligned}
\tilde{\delta}_{mn} &= \left(\frac{1}{r} \sum_{i=0}^{r-1} \cos\left(\frac{2\pi i}{r} m\right) \cos\left(\frac{2\pi i}{r} n\right)\right) + \cos(2\pi m)\cos(2\pi n) \\
&\quad + \left(\frac{1}{r} \sum_{i=1}^{r} \cos\left(\frac{2\pi i}{r} m\right) \cos\left(\frac{2\pi i}{r} n\right)\right) + \cos(0)\cos(0) \\
&\quad - \frac{1}{r}\left(\cos(0)\cos(0) + \cos(2\pi m)\cos(2\pi n)\right) \\
&= \left(\frac{1}{r} \sum_{i=0}^{r-1} \cos\left(\frac{2\pi i}{r} m\right) \cos\left(\frac{2\pi i}{r} n\right)\right) + \cos\left(\frac{2\pi r}{r} m\right)\cos\left(\frac{2\pi r}{r} n\right) \\
&\quad + \left(\frac{1}{r} \sum_{i=1}^{r} \cos\left(\frac{2\pi i}{r} m\right) \cos\left(\frac{2\pi i}{r} n\right)\right) + \cos(0)\cos(0) \\
&\quad - \frac{1}{r}\left(\cos(0)\cos(0) + \cos(2\pi m)\cos(2\pi n)\right) \\
&= \frac{2}{r} \sum_{i=0}^{r} \left(\cos\left(\frac{2\pi i}{r} m\right) \cos\left(\frac{2\pi i}{r} n\right)\right) - \frac{1}{r}\left(\cos(0)\cos(0) + \cos(2\pi m)\cos(2\pi n)\right)
\end{aligned}$$

Since $m$ and $n$ are integers:

$$\tilde{\delta}_{mn} = \frac{2}{r} \sum_{i=0}^{r} \left(\cos\left(\frac{2\pi i}{r} m\right) \cos\left(\frac{2\pi i}{r} n\right)\right) - \frac{2}{r}. \tag{19}$$

In fact, in the limit, $r \to \infty$, only the first term in the right hand side of Equation 19 matters in our approximation. Let

$$\hat{\delta}_{mn} \doteq \frac{2}{r} \sum_{i=0}^{r} \left(\cos\left(\frac{2\pi i}{r} m\right) \cos\left(\frac{2\pi i}{r} n\right)\right), \tag{20}$$

we then have

$$\lim_{r\to\infty} \tilde{\delta}_{mn} = \lim_{r\to\infty} \hat{\delta}_{mn} - \lim_{r\to\infty} \frac{2}{r} = \lim_{r\to\infty} \hat{\delta}_{mn} - 0, \tag{21}$$

and consequently

$$\lim_{r\to\infty} \hat{\delta}_{mn} = \delta_{mn}. \tag{22}$$

### B.1.1 Using The Kronecker Delta Function to approximate GaLiTe

We start with the GaLiTe recurrent state update which we will then approximate using the Kronecker delta approximation introduced above. GaLiTe recurrent state update is expressed as follows:

$$\mathbf{C}_t = \big((1 - \beta_t) \otimes (1 - \gamma_t)\big) \odot \mathbf{C}_{t-1} + \big(\beta_t \odot \mathbf{v}_t\big) \otimes \big(\gamma_t \odot \mathbf{k}_t\big). \tag{23}$$

We use the approximation of the Kronecker delta function in Equation 20 to approximate the update in Equation 23. We will start by representing the recurrent state $\mathbf{C}_t$ as a sum of outer products. We will do so by recursively expanding using the definition of $\mathbf{C}_t$. Following the recursive definition we have:

$$
\begin{aligned}
\mathbf{C}_t &= \big((1 - \beta_t) \otimes (1 - \gamma_t)\big) \odot \mathbf{C}_{t-1} + \big(\beta_t \odot \mathbf{v}_t\big) \otimes \big(\gamma_t \odot \mathbf{k}_t\big) \\
&= \Big((\beta_t \odot \mathbf{v}_t) \otimes (\gamma_t \odot \mathbf{k}_t)\Big) \\
&\quad + \Big((1 - \beta_t) \otimes (1 - \gamma_t)\Big) \odot \Big((\beta_{t-1} \odot \mathbf{v}_{t-1}) \otimes (\gamma_{t-1} \odot \mathbf{k}_{t-1}) + \big((1 - \beta_{t-1}) \otimes (1 - \gamma_{t-1})\big) \odot \mathbf{C}_{t-2}\Big) \\
&= \Big((\beta_t \odot \mathbf{v}_t) \otimes (\gamma_t \odot \mathbf{k}_t)\Big) + \Big((1 - \beta_t) \otimes (1 - \gamma_t)\Big) \odot \Big((\beta_{t-1} \odot \mathbf{v}_{t-1}) \otimes (\gamma_{t-1} \odot \mathbf{k}_{t-1})\Big) \\
&\quad + \Big((1 - \beta_t) \otimes (1 - \gamma_t)\Big) \odot \Big((1 - \beta_{t-1}) \otimes (1 - \gamma_{t-1})\Big) \odot \mathbf{C}_{t-2}
\end{aligned}
$$

Using the fact that $(\mathbf{a} \otimes \mathbf{b}) \odot (\mathbf{c} \otimes \mathbf{d}) = (\mathbf{a} \odot \mathbf{c}) \otimes (\mathbf{b} \odot \mathbf{d})$ for arbitrary vectors $\mathbf{a}$, $\mathbf{b}$, $\mathbf{c}$, $\mathbf{d}$, we have:

$$
\begin{aligned}
\mathbf{C}_t &= \Big((\beta_t \odot \mathbf{v}_t) \otimes (\gamma_t \odot \mathbf{k}_t)\Big) + \Big(\big((1 - \beta_t) \odot \beta_{t-1} \odot \mathbf{v}_{t-1}\big) \otimes \big((1 - \gamma_t) \odot \gamma_{t-1} \odot \mathbf{k}_{t-1}\big)\Big) \\
&\quad + \Big(\big((1 - \beta_t) \odot (1 - \beta_{t-1})\big) \otimes \big((1 - \gamma_t) \odot (1 - \gamma_{t-1})\big)\Big) \odot \mathbf{C}_{t-2}
\end{aligned}
$$

Recursively expanding it further and regrouping the terms similar to above:

$$
\begin{aligned}
&= \Big((\beta_t \odot \mathbf{v}_t) \otimes (\gamma_t \odot \mathbf{k}_t)\Big) + \Big(\big((1 - \beta_t) \odot \beta_{t-1} \odot \mathbf{v}_{t-1}\big) \otimes \big((1 - \gamma_t) \odot \gamma_{t-1} \odot \mathbf{k}_{t-1}\big)\Big) + \\
&\quad + \Big(\big((1 - \beta_t) \odot (1 - \beta_{t-1}) \odot \beta_{t-1} \odot \mathbf{v}_{t-2}\big) \otimes \big((1 - \gamma_t) \odot (1 - \gamma_{t-1}) \odot \gamma_{t-2} \odot \mathbf{k}_{t-2}\big)\Big) + \mathbf{C}_{t-3}
\end{aligned}
$$

Next, we rewrite the recursive expansion as a single summation term of outer product. This is possible because upon following the recursive definition, we can see above that each term of the expansion could be written as an outer product of vectors. The vectors used to calculate the outer product at each timestep consists of element-wise vector product involving either the value or key, and their corresponding gating vectors. We show the expansion for only until $t - 3$, but the same steps could be followed until $t = 0$. We define product operation $\prod$ such that: $\prod_i^j(\mathbf{a}_i) \doteq 1$ if $j > i$, and $\prod_i^j(\mathbf{a}_i) \doteq \mathbf{a}_i \odot \mathbf{a}_{i+1} \odot \ldots \mathbf{a}_j$. We introduce variables $\mathbf{l}_i$ and $\mathbf{m}_i$ to represent the left and right terms of the outer-product at a given timestep,

for $i = 0, 1, \ldots, t$. We then define $\mathbf{C}_t$ in terms of these variables to simplify the equation above:

$$\mathbf{C}_t = \sum_{i=0}^{t} \mathbf{l}_i \otimes \mathbf{m}_i \tag{24}$$

$$\mathbf{l}_i = \prod_{j=i+1}^{t} (1 - \beta_j) \odot \beta_i \odot \mathbf{v}_i \tag{25}$$

$$\mathbf{m}_i = \prod_{j=i+1}^{t} (1 - \gamma_j) \odot \gamma_i \odot \mathbf{k}_i \tag{26}$$

Next, we use the approximate Kronecker delta function in Equation 20 to approximate the sum of outer products in Equation 24. We start by introducing the Kronecker delta function $\delta_{mn}$ into the expression of $\mathbf{C}_t$ above by introducing a double summation:

$$\mathbf{C}_t = \sum_{i=0}^{t} \mathbf{l}_i \otimes \mathbf{m}_i = \sum_{j=0}^{t} \sum_{i=0}^{t} \delta_{ij} \mathbf{l}_i \otimes \mathbf{m}_j$$

Then we replace the Replacing $\delta_{i,j}$ with $\hat{\delta}_{i,j}$ we obtain an approximation $\tilde{\mathbf{C}}_t$ of $\mathbf{C}_t$ as follows:

$$\begin{aligned} \mathbf{C}_t \approx \tilde{\mathbf{C}}_t &= \sum_{j=0}^{t} \sum_{i=0}^{t} \hat{\delta}_{ij} \mathbf{l}_i \otimes \mathbf{m}_j \\ &= \frac{2}{r} \sum_{j=0}^{t} \sum_{i=0}^{t} \sum_{k=0}^{r} \cos\left(\frac{2\pi k}{r} i\right) \cos\left(\frac{2\pi k}{r} j\right) \mathbf{l}_i \otimes \mathbf{m}_j \\ &= \frac{2}{r} \sum_{k=0}^{r} \sum_{j=0}^{t} \sum_{i=0}^{t} \cos\left(\frac{2\pi k}{r} i\right) \cos\left(\frac{2\pi k}{r} j\right) \mathbf{l}_i \otimes \mathbf{m}_j, \end{aligned}$$

where we first applied Equation 20 followed by rearranging the order of the summations. Let $\omega_k \doteq \cos\left(\frac{2\pi k}{r}\right)$, we then have:

$$\tilde{\mathbf{C}}_t = \frac{2}{r} \sum_{k=0}^{r} \sum_{j=0}^{t} \sum_{i=0}^{t} \cos\left(\omega_k i\right) \cos\left(\omega_k j\right) \mathbf{l}_i \otimes \mathbf{m}_j.$$

Because $(ab)(\mathbf{c} \otimes \mathbf{d}) = (a\mathbf{c}) \otimes (b\mathbf{d})$ for scalars $a, b$ and vectors $\mathbf{c}, \mathbf{d}$, this allows us to seperate the two cosine terms in the summation:

$$\tilde{\mathbf{C}}_t = \frac{2}{r} \sum_{k=0}^{r} \sum_{j=0}^{t} \sum_{i=0}^{t} \left(\cos\left(\omega_k i\right) \mathbf{l}_i\right) \otimes \left(\cos\left(\omega_k j\right) \mathbf{m}_j\right).$$

Next, we will use the distributive property: $\left(\sum_{i=0}^{m} \mathbf{a}_i\right) \otimes \left(\sum_{j=0}^{n} \mathbf{b}_j\right) = \sum_{i=0}^{m} \sum_{j=0}^{n} (\mathbf{a}_i \otimes \mathbf{b}_j)$, where $\mathbf{a}_0, \ldots, \mathbf{a}_m$ and $\mathbf{b}_0, \ldots, \mathbf{b}_n$ are arbitrary vectors. We will use it to rewrite the above equation as a single summation of outer products, where the summation is defined only over $r$. This is a key operation that allows us to write the approximation to be written temporally iterative manner. Applying the distributive property from right

to left into the equation above, and then using Equations 25 and 26, we have:

$$\tilde{\mathbf{C}}_t = \frac{2}{r} \sum_{k=0}^{r} \sum_{j=0}^{t} \sum_{i=0}^{t} \left( \cos\left(\omega_k i\right) \mathbf{l}_i \right) \otimes \left( \cos\left(\omega_k j\right) \mathbf{m}_j \right).$$

$$= \frac{2}{r} \sum_{k=0}^{r} \left( \sum_{i=0}^{t} \cos\left(\omega_k i\right) \mathbf{l}_i \right) \otimes \left( \sum_{i=0}^{t} \cos\left(\omega_k i\right) \mathbf{m}_i \right)$$

$$= \frac{2}{r} \sum_{k=0}^{r} \left( \sum_{i=0}^{t} \cos\left(\omega_k i\right) \prod_{j=i+1}^{t} \left(1 - \beta_j\right) \odot \beta_i \odot \mathbf{v}_i \right) \otimes \left( \sum_{i=0}^{t} \cos\left(\omega_k i\right) \prod_{j=i+1}^{t} \left(1 - \gamma_j\right) \odot \gamma_i \odot \mathbf{k}_i \right). \quad (27)$$

Next, we simplify the above equation and rewrite it in a recurrent form. Let $\tilde{\mathbf{v}}_t^k$ and $\tilde{\mathbf{k}}_t^k$ be defined as:

$$\tilde{\mathbf{v}}_t^k \doteq \sum_{i=0}^{t} \cos\left(\omega_k i\right) \prod_{j=i+1}^{t} \left(1 - \beta_j\right) \odot \beta_i \odot \mathbf{v}_i, \quad (28)$$

$$\tilde{\mathbf{k}}_t^k \doteq \sum_{i=0}^{t} \cos\left(\omega_k i\right) \prod_{j=i+1}^{t} \left(1 - \gamma_j\right) \odot \gamma_i \odot \mathbf{k}_i. \quad (29)$$

$\tilde{\mathbf{C}}_t$ could then then written in terms of $\tilde{\mathbf{v}}_t^k$ and $\tilde{\mathbf{k}}_t^k$ as:

$$\tilde{\mathbf{C}}_t = \frac{2}{r} \sum_{k=0}^{r} \tilde{\mathbf{v}}_t^k \otimes \tilde{\mathbf{k}}_t^k, \quad (30)$$

We regroup the terms and derive a recursive relationship of $\tilde{\mathbf{v}}_t^k$ and $\tilde{\mathbf{k}}_t^k$ with respect to $\tilde{\mathbf{v}}_{t-1}^k$ and $\tilde{\mathbf{k}}_{t-1}^k$. First, we separate the term in the summation when $i = t$. Next, we factorize $(1 - \beta_t)$ from the summation term. Finally, following the definition in Equation 28, we replace the second term with $\tilde{\mathbf{v}}_{t-1}^i$.

$$\tilde{\mathbf{v}}_t^k = \sum_{i=0}^{t} \cos(\omega_k i) \prod_{j=i+1}^{t} \left(1 - \beta_j\right) \odot \beta_i \odot \mathbf{v}_i$$

$$= \cos(\omega_k t)\beta_t \odot \mathbf{v}_t + \sum_{i=0}^{t-1} \cos(\omega_k i) \prod_{j=i+1}^{t} \left(1 - \beta_j\right) \odot \beta_i \odot \mathbf{v}_i$$

$$= \cos(\omega_k t)\beta_t \odot \mathbf{v}_t + (1 - \beta_t) \sum_{i=0}^{t-1} \cos(\omega_k i) \prod_{j=i+1}^{t-1} \left(1 - \beta_j\right) \odot \beta_i \odot \mathbf{v}_i$$

$$= \cos(\omega_k t)\beta_t \odot \mathbf{v}_t + (1 - \beta_t) \odot \tilde{\mathbf{v}}_{t-1}^k \quad (31)$$

Similarly, we apply the same operations to $\tilde{\mathbf{k}}_t^i$:

$$\tilde{\mathbf{k}}_t^k = \sum_{i=0}^{t} \cos(\omega_k i) \prod_{j=i+1}^{t} \left(1 - \gamma_j\right) \odot \gamma_i \odot \mathbf{k}_i$$

$$= \cos(\omega_k t)\gamma_t \odot \mathbf{k}_t + \sum_{i=0}^{t-1} \cos(\omega_k i) \prod_{j=i+1}^{t} \left(1 - \gamma_j\right) \odot \gamma_i \odot \mathbf{k}_i$$

$$= \cos(\omega_k t)\gamma_t \odot \mathbf{k}_t + (1 - \gamma_t) \sum_{i=0}^{t-1} \cos(\omega_k i) \prod_{j=i+1}^{t-1} \left(1 - \gamma_j\right) \odot \gamma_i \odot \mathbf{k}_i$$

$$= \cos(\omega_k t)\gamma_t \odot \mathbf{k}_t + (1 - \gamma_t) \odot \tilde{\mathbf{k}}_{t-1}^k \quad (32)$$

Using the recursive relationships in Equation 31 and 32, we can now present the final approximation. For a given $r$, we maintain recurrent states $\tilde{\mathbf{v}}_{t-1}^k$ and $\tilde{\mathbf{k}}_{t-1}^k$ for $k = 0, 1, 2, \ldots, r$. For $\omega_k \doteq \frac{2\pi k}{r}$, and assuming $\tilde{\mathbf{v}}_0^i$ and $\tilde{\mathbf{k}}_0^i$ are initialized as zeros, the recurrent updates to $\tilde{\mathbf{v}}_t^i$ and $\tilde{\mathbf{k}}_t^i$ and further the approximation to $\mathbf{C}_t$ are given by:

$$\mathbf{C}_t \approx \tilde{\mathbf{C}}_t = \frac{2}{r} \sum_{k=0}^r \tilde{\mathbf{v}}_{\mathbf{t}}^{\mathbf{k}} \otimes \tilde{\mathbf{k}}_t^k \tag{33}$$

where, for $k = 0, 1, 2, \ldots, r$ we have:

$$\tilde{\mathbf{v}}_t^k \doteq \cos(\omega_k t)\beta_t \odot \mathbf{v}_t + (1 - \beta_t) \odot \tilde{\mathbf{v}}_{t-1}^k \tag{34}$$

$$\tilde{\mathbf{k}}_t^k \doteq \cos(\omega_k t)\gamma_t \odot \mathbf{k}_t + (1 - \gamma_t) \odot \tilde{\mathbf{k}}_{t-1}^k \tag{35}$$

Since $\lim_{r \to \infty} \hat{\delta}_{mn} = \delta_{mn}$, it follows that $\lim_{r \to \infty} \tilde{\mathbf{C}}_t = \mathbf{C}_t$. Unlike Equation 23, Equation 34 and 35 define a recurrence over vectors instead of matrices, and if $r \ll d$, the recurrence is much more efficient in space than the recurrence in Equation 23. We leave it to future work to formally derive the approximation error. In Section E we show the approximation error with a synthetic error under different values of $r$.

Lastly, since the current state $\tilde{\mathbf{C}}_t$ could be represented as a sum of outer products in a non-recurrent manner, we can avoid explicitly calculating $\tilde{\mathbf{C}}_t$ and instead calculate the attention output $\mathbf{a}_t$ as follows:

$$\mathbf{a}_t \doteq \frac{\sum_{k=0}^r \tilde{\mathbf{v}}_t^k \left( \left( \tilde{\mathbf{k}}_t^k \right)^\top \mathbf{q}_t \right)}{2r(\mathbf{s}_t^\top \mathbf{q}_t)} \tag{36}$$

# C   Approximate Gated Linear Transformer (AGaLiTe)

Algorithm 4 shows the Approximate Gated Linear Transformer (AGaLiTe). We highlight changes from Algorithm 3 in blue. The algorithm maintains a set of vectors $\tilde{\mathbf{k}}_{t-1}^0, ..., \tilde{\mathbf{k}}_{t-1}^r \in \mathbb{R}^{\eta d_h}$, $\tilde{\mathbf{v}}_{t-1}^0, ..., \tilde{\mathbf{v}}_{t-1}^r \in \mathbb{R}^{d_h}$, and $\mathbf{s}_{t-1} \in \mathbb{R}^{\eta d_h}$ as the recurrent state at a given time-step $t$. The number of vectors stored could be controlled by modifying the hyperparameter $r$, which should ideally be set to a small value. The key, query, and value vectors are calculated similarly to GaLiTe. The recurrent state update is modified to use the approximation in Equation 33. At each time step, the recurrent vectors are updated using element-wise vector multiplication and addition operations (lines 10-14). The operation on each recurrent vector could be executed in parallel. The attention output is calculated without ever explicitly calculating $\tilde{\mathbf{C}}_t$ (lines 16-18).

---

**Algorithm 4** Approximate Gated Linear Transformer (AGaLiTe) Self-Attention (Streaming Data)

---

**Input**: $\mathbf{x}_t \in \mathbb{R}^d$, $\tilde{\mathbf{k}}_{t-1}^0, ..., \tilde{\mathbf{k}}_{t-1}^r \in \mathbb{R}^{\eta d_h}$, $\tilde{\mathbf{v}}_{t-1}^0, ..., \tilde{\mathbf{v}}_{t-1}^r \in \mathbb{R}^{d_h}$, and $\mathbf{s}_{t-1} \in \mathbb{R}^{\eta d_h}$
**Hyperparameters**: $\eta$ and $r$.
**Parameters**: $\mathbf{W}_K, \mathbf{W}_Q, \mathbf{W}_V, \mathbf{W}_\beta, \mathbf{W}_\gamma \in \mathbb{R}^{d_h \times d}$ and $\mathbf{W}_{p_1}, \mathbf{W}_{p_2}, \mathbf{W}_{p_3} \in \mathbb{R}^{\eta \times d}$

1: Assume $\mathbf{s}_0 \leftarrow 0, \mathbf{C}_0 \leftarrow 0$.

                                                          {Calculate Key}

2: $\mathbf{k}_t \leftarrow f(relu(\mathbf{W}_{p_1}\mathbf{x}_t) \otimes relu(\mathbf{W}_K\mathbf{x}_t))$

                                                         {Calculate Query}

3: $\mathbf{q}_t \leftarrow f(relu(\mathbf{W}_{p_2}\mathbf{x}_t) \otimes relu(\mathbf{W}_Q\mathbf{x}_t))$

                                                        {Calculate Value}

4: $\mathbf{v}_t \leftarrow \mathbf{W}_V\mathbf{x}_t$

                                           {Generate Gating Vectors}

5: $\beta_t \leftarrow \sigma_g(\mathbf{W}_\beta\mathbf{x}_t)$
6: $\gamma_t \leftarrow f(\sigma_g(\mathbf{W}_{p_3}\mathbf{x}_t) \otimes \sigma_g(\mathbf{W}_\gamma\mathbf{x}_t))$

                                               {Update Memory}

7: **for** $i \leftarrow 0$ to $r$ **in parallel, do**
8:     $\omega_i \leftarrow (2\pi i)/r$
9:     $\tilde{\mathbf{v}}_t^i \leftarrow \tilde{\mathbf{v}}_{t-1}^i \odot (1 - \beta_t) + \cos(\omega_i t)(\beta_t \odot \mathbf{v}_t)$
10:    $\tilde{\mathbf{k}}_t^i \leftarrow \tilde{\mathbf{k}}_{t-1}^i \odot (1 - \gamma_t) + \cos(\omega_i t)(\gamma_t \odot \mathbf{k}_t)$
11: **end for**
12: $\mathbf{s}_t \leftarrow (1 - \gamma_t) \odot \mathbf{s}_{t-1} + \gamma_t \odot \mathbf{k}_t$

                                  {Calculate Attention Vector}

13: $\mathbf{a} \leftarrow \sum_{i=0}^r \tilde{\mathbf{v}}_t^i (\tilde{\mathbf{k}}_t^{i\top}\mathbf{q}_t)$
14: $\mathbf{b} \leftarrow 2r(\mathbf{s}_t^\top\mathbf{q}_t)$
15: $\mathbf{a}_t \leftarrow \mathbf{a}/\mathbf{b}$
**Output**: $\mathbf{a}_t \in \mathbb{R}^{d_h}$, $\tilde{\mathbf{k}}_t^0, ..., \tilde{\mathbf{k}}_t^r \in \mathbb{R}^{\eta d_h}$, $\tilde{\mathbf{v}}_t^0, ..., \tilde{\mathbf{v}}_t^r \in \mathbb{R}^{d_h}$, and $\mathbf{s}_t \in \mathbb{R}^{\eta d_h}$

---

# D   Results in Long Range Arena

We additionally evaluate on the ListOps and Text (IMDB) from the Long Range Arena (Tay et al., 2021) as to evaluate the architecture's ability to learn long-range dependencies in a supervised learning scenario. The performance of AGaLiTe ($\eta = 8, r = 1$) is compared with the transformer's and linear transformer's in Table 2. The exact hyperparameters for AGaLiTe are listed in Table 4. We found that AGaLiTe outperforms the previously reported results of the transformer and linear transformer architecture (Tay et al., 2021) in both of these tasks.

Table 2: Results in Long Range Arena. Reported as mean ± std error over 10 seeds.

.

(a) ListOps Task

| Model | Params | Score |
|---|---|---|
| Linear Transformer | 8.9M | 16.13 |
| Transformer | 8.9M | 36.37 |
| AGaLiTe | 1.7M | **39.33 ± 0.34** |

(b) Text (IMDB) Task

| Model | Params | Score |
|---|---|---|
| Linear Transformer | 3.4M | 65.9 |
| Transformer | 3.4M | 64.27 |
| AGaLiTe | 1.7M | **79.83 ± 1.3** |

## E   Effect of $r$ on the Quality of Approximation in AGaLiTe

We empirically evaluate the effect of $r$ on the quality of the approximation of the current state matrix $\mathbf{C}_t$. Ideally, we want to set $r$ to a small value as the space complexity of AGaLiTe is directly proportional to $r$. We consider a synthetic example where the value $\mathbf{v}_t$ and key $\mathbf{k}_t$ at each time step are sampled randomly from a normal distribution. We set the embedding dimension $d$ to 128 and randomly sample values and keys for 100 timesteps. Instead of using vectors $\gamma_t$ and $\beta_t$ for gating at every timestep, we use a constant value $c$. We then compare the difference between the current state matrix $\mathbf{C}_t$ computed using the exact method in Equation 5, with the current state matrix $\tilde{\mathbf{C}}_t$ computed using the approximate method in Equation 33 at the 100th time-step. We use the Frobenius norm to measure the difference between the two matrices. We repeat the experiment for different values of $r$ and $c$. For each configuration, we report the mean error across 50 independent runs. Figure 7 shows the results of this experiment. We observe that the error in approximation decreases with increasing value of $r$. For most values of $r$ and $c$, the approximation error is low. This is useful since it allows us to set $r$ to a small value, thereby reducing the space complexity of the model. In fact, in the largest experiments described in this thesis, we set $r$ to 7. Interestingly, we observe periodic bands in the error plot. It is possible that this is due to the periodicity of the cosine functions used in the attention mechanism. We leave further exploration around the theoretical nature of the error in approximation for future work.

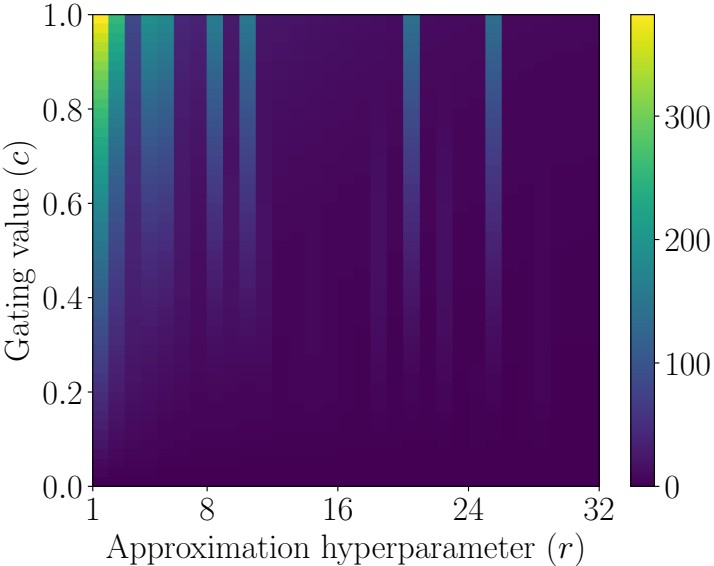

Figure 7: Error in approximating the current state $\mathbf{C}_t$ for different values $r$ and gating at $t = 100$ for randomly sampled values and keys.

# F    Parallelization over an Input Sequence

Transformers are naturally designed for parallelism over a sequence of input data, as the self-attention operation does not have dependencies between different parts of the input sequence. It is essential to consider the parallelizability of transformer architectures, when the input sequence is presented in a batched fashion. Such a scenario is common in practice, as most existing actor-critic approaches such as PPO and A2C (Schulman et al., 2017; Mnih et al., 2016) estimate gradient updates to the actor and critic using batches of trajectories collected through agent-environment interactions. Furthermore, most modern hardware accelerators, such as GPUs and TPUs, excel in handling parallelizable algorithms, and parallelization is vital for effectively training large models.

Extension of Algorithm 3 and 4 to accommodate parallelization over a sequence of inputs is straightforward, depending on whether the computation has dependencies on the previous state or not. The majority of the computations in both algorithms, which involve calculating keys, queries, values, gating vectors, and the attention vector, do not depend on the previous state and can be parallelized over the sequence. The only part of the algorithm that depends on the previous state is the update of the current state. In Algorithm 3, this is done from lines 13-14, and in Algorithm 4, from lines 10-15. The update of the current state in both algorithms is implemented as a first order recurrence. This operation is parallelizable as such recurrences could be expressed as an associtiave binary operations (see Blelloch, 1990). In our implementation, we used the *associative_scan* operation in Jax to parallelize GaLiTe and AGaLiTe over an input sequence.

# G Additional Experiment Details

## G.1 T-Maze

**Environment Description:** The T-Maze environment considered in this paper is similar to the one proposed by Bakker (2001). Figure 8 shows two possible episodes in the T-Maze environment. At each timestep, the agent receives a 16-bit binary observation. The first two bits correspond to the cue signal which is either 01 or 10 at the first timestep of an episode, depending on whether the reward is located at the left or right turn at the intersection, respectively. The cue bits are zero in all other timesteps. We consider the largest possible corridor length as 200. To encode the corridor information, the agent additionally receives 8-bit gray code encoding of its current location. The gray code encoding is zero at the beginning of an episode and is updated at each timestep. To make the problem more challenging, we added 6 noisy distractor bits to the observation. The distractor bits are sampled uniformly at random at each timestep. The agent can take one of the four possible discrete actions at each timestep: up, down, left, or right. The agent receives a reward of -0.1 at each non-terminal timestep. At termination, the agent receives a reward of +4 for taking the correct turn and a reward of -1 for taking an incorrect turn. The reward of +4 is chosen to encourage the agent to take the correct turn at the intersection. The difficulty of this environment can be increased by increasing the corridor length. Increasing the corridor length requires the agent to remember the signal for a longer number of timesteps. Since the agent's observations include distractor bits, the agent also needs to learn to ignore the distractor bits and focus on the cue signal.

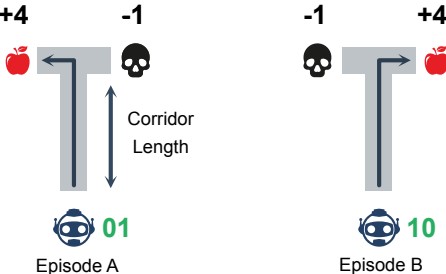

Figure 8: The T-Maze environment. The agent has to remember a binary cue (denoted by green text), shown only at the beginning of the episode, in order to take the correct turn at the intersection and receive a positive reward. The figure shows two possible episodes and the optimal path an agent must take. The agent's current location is provided as gray code encoding in the observation, along with distractor signals. The corridor length could be varied to increase the difficulty of the problem.

**Hyperparameters and Tuning Strategy:** We include the architecture configuration for each of the 5 architectures in Table 4. Our hyperparameter tuning strategy is as follows: We train 5 seeds per architecture for each corridor length in 120-200 and hyperparameter configuration for 5M steps. We identify the best hyperparameter configuration according to the best mean success rate in the last 100K steps across all corridor lengths.

A few additional details are worth reporting for the purposes of reproducibility. We conducted all experiments using Python and implemented the agents using the Jax library (Bradbury et al. (2018)). We used the GTrXL implementation from the DIEngine library (engine Contributors, 2021). Each agent is trained using 16-core machine with 12GB RAM. The network weights are initialized using orthogonal initialization (Saxe et al. (2014)). A single run using the slowest architecture takes around 20 hours to complete.

## G.2 Partially Observable CartPole

Table 5 shows PPO hyperparameters used for CartPole experiments. We show additional results for the partially observable CartPole environment when no noise is added to the observation vector in figure 9

Table 3: Hyperparameters and sweeps for the T-Maze experiments.

| Hyperparameter | Value |
|---|---|
| Learning Rate | [0.001, 0.0001 0.0005, 0.00001, 0.00005] |
| Discount Factor ($\gamma$) | 0.99 |
| Advantage Estimation Coefficient ($\lambda$) | 0.95 |
| Entropy Coefficient | [0.1, 0.01, 0.001, 0.0001, 0.00001] |
| Value Loss Coefficient | 0.5 |
| Rollout Len | 256 |
| Num of Envs | 8 |
| Batch Size (Rollout Len × Num of Envs) | 2048 |
| Actor Layer Dimension | 128 |
| Critic Layer Dimension | 128 |

Table 4: Architecture configuration for LSTM, GRU, GTrXL, GaLiTe, and AGaLiTe for T-Maze, Mystery-Path experiments, and Craftax experiments.

| Hyperparameter | LSTM | GRU | GTrXL | GaLiTe | AGaLiTe |
|---|---|---|---|---|---|
| Embedding Dimension ($d$) | 600 | 680 | 128 | 128 | 128 |
| Hidden Dimension | 1200 | 1360 | N/A | N/A | N/A |
| Num Heads | N/A | N/A | 4 | 4 | 4 |
| Head Dim ($d_h$) | N/A | N/A | 64 | 64 | 64 |
| Num Layers ($L$) | 1 | 1 | 4 | 4 | 4 |
| Memory Size ($M$) | N/A | N/A | [128, 256] | N/A | N/A |
| Projection Hyperparameter ($\eta$) | N/A | N/A | N/A | 4 | [4,8] |
| Approximation Hyperparameter ($r$) | N/A | N/A | N/A | N/A | 1 |
| Actor Layer Dimension | 128 | - | - | - | - |
| Critic Layer Dimension | 128 | - | - | - | - |

Table 5: Hyperparameters and sweeps for the CartPole experiments.

| Hyperparameter | Value |
|---|---|
| Learning Rate | [0.01, 0.001, 0.0001, 0.00001] |
| Discount Factor ($\gamma$) | 0.99 |
| Advantage Estimation Coefficient ($\lambda$) | 0.9 |
| Entropy Coefficient | 0.0 |
| Value Loss Coefficient | 1.0 |
| Rollout Len | 1024 |
| Num of Envs | 1 |
| Batch Size (Rollout Len × Num of Envs) | 1024 |
| Number of Epochs | 10 |
| PPO Clip Ration | 0.2 |
| Max Gradient Norm | 0.5 |

### G.3 Mystery Path

**Environment Description:** Pleines et al. ( 2023) introduced the Mystery Path environment as part of the Memory Gym benchmark, which aimed to test agents' abilities to memorize many events over an episode. The Mystery Path is a $7 \times 7$ grid environment with pixel-based observations. At the beginning of each episode, the start position of the agent, the origin, is sampled from the grid's borders. Then, the target position is sampled from the grid's borders on the opposite side of the origin. A randomly generated path then connects both the origin and the goal. Figure 10a shows an example of a generated origin, goal, and path. The agent's observation, shown in Figure 10b, is a $64 \times 64$ RGB image containing the origin,

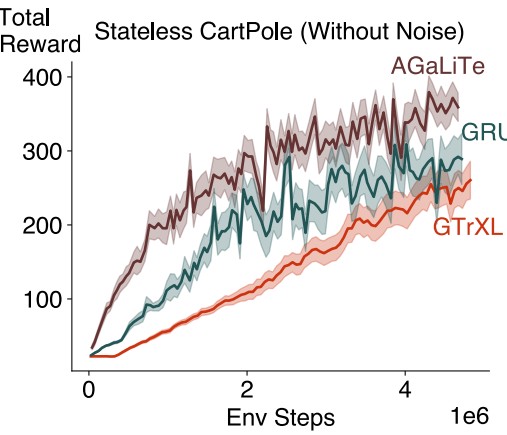

Figure 9: Non-noisy Partially Observable CartPole

the target, and the agent. The agent gets a +1 reward when it reaches the goal and a 0.1 reward when visiting a new tile on the path to the goal. If the agent falls off the path, as in Figure 10c, a red cross appears as visual feedback, and the agent returns to the origin. The reward is zero in all other timesteps. We consider two variants of this environment, MPGrid and MP. MPGrid has maximum episode length of 128, uses grid-like movements and 4 possible actions (left, right, up and down). On the other hand, MP has a maximum episode length of 512, has smoother movements, and a larger action space that allows diagonal movements.

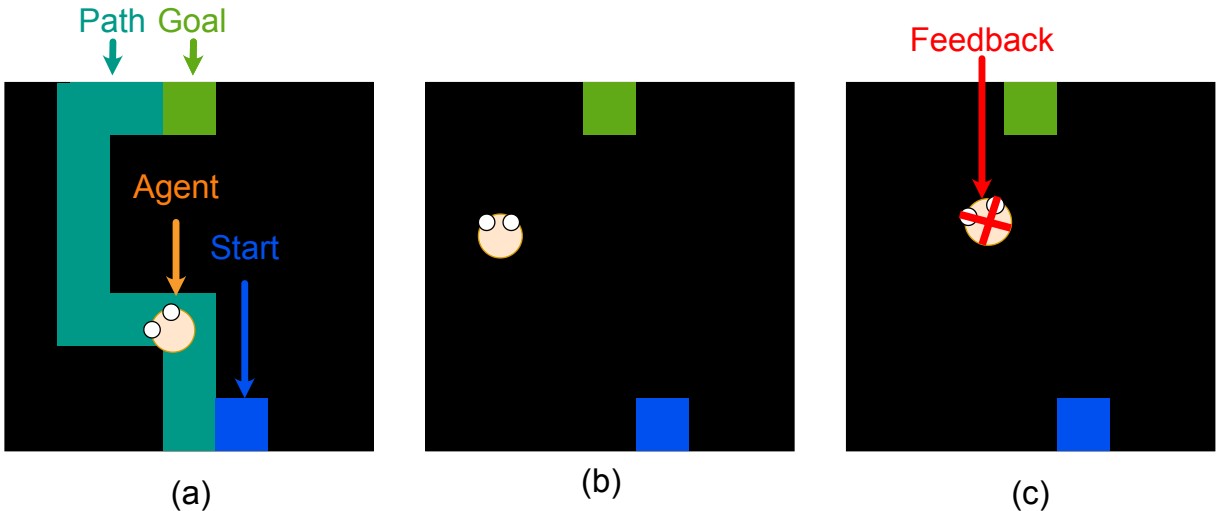

Figure 10: A visualization of the Mystery Path environment.

**Hyperparameters and Tuning Strategy:** The architecture sizes used for Mystery Path experiments are kept same as in Table 4, however, we used actor and critic layer dimension of 256. We detail the hyperparameters used for the PPO algorithm that used for training the agents in the Mystery Path environment in Table 6. We tune learning rate and entropy coefficient for the sweeps mentioned in Table 6. Our hyperparameter tuning strategy is as follows: we train 3 seeds per architecture for each the hyperparameter configuration for 60M steps in the Mystery Path Grid environment. Finally, we identify the best hyperparameter configuration according to the best episodic reward in the last 1M training steps.

Table 6: Hyperparameters and sweeps for Mystery Path experiments.

| Hyperparameter | Value |
|---|---|
| Learning Rate | [0.0025, 0.00025, 0.000025] |
| Discount Factor ($\gamma$) | 0.99 |
| Advantage Estimation Coefficient ($\lambda$) | 0.95 |
| Entropy Coefficient | [0.03, 0.003, 0.0003, 0.00003] |
| Number of Epochs | 3 |
| Rollout Length | 128 |
| Sequence Length | 128 |
| Number of Env | 128 |
| Batch Size (Sequence Length $\times$ Number of Env) | 16384 |
| Number of Mini Batches | 8 |
| PPO Clip Ratio | 0.2 |
| Max Gradient Norm | 4 |
| Value Function Coefficient | 0.5 |

## G.4 Memory Maze

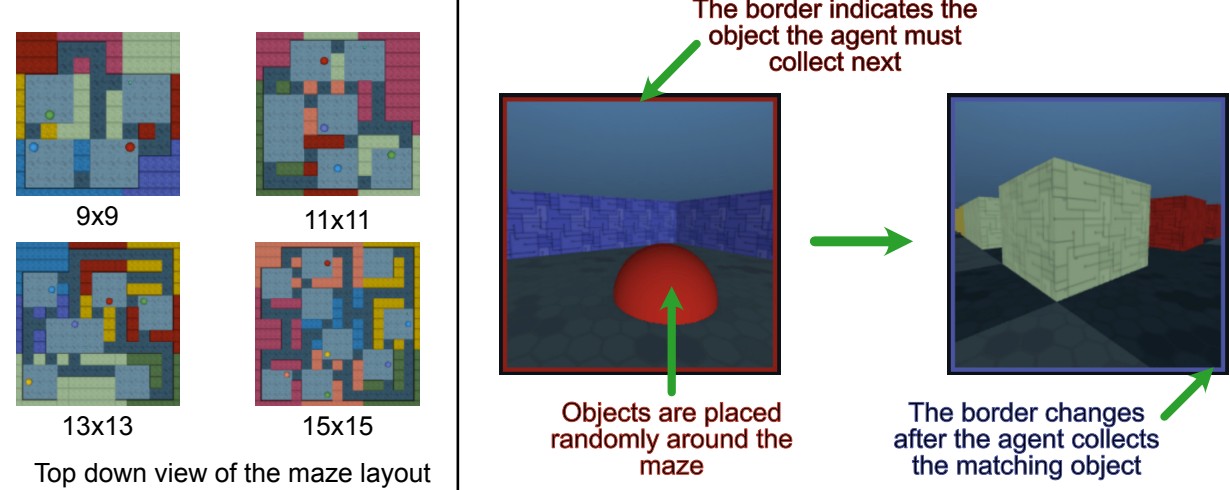

Figure 11: The Memory Maze environment. On the left, we show a possible maze layout for all four Memory Maze configurations. The maze layout is randomized at each episode. On the right, we show two sample observations that the agent receives. The agent's observation at each time-step is $64 \times 64$ RGB pixels and the action space is discrete. The border color of the observation image indicates the target object color which the agent needs to find to receive a reward. After collecting the object, the border color changes, indicating the next target object. The episode lengths are fixed depending on the Memory Maze configuration, with larger configurations having longer episodes.

**Environment Description:** The Memory Maze environment evaluates an agent's long-term memory capabilities in a partially observable RL setting. Figure 11 illustrates this environment. The agent's observation at each time-step is an image with $64 \times 64$ RGB pixels, and the action space is discrete. In each episode, the agent starts in a randomly generated maze containing several objects of different colors. The agent's objective is to find the target object of a specific color, indicated by the border color in the observation image. Upon successfully touching the correct object, the agent receives a +1 reward, and the next random object is chosen as the new target. If the agent touches an object of the wrong color, there is no effect on the environment. The maze layout and object locations remain constant throughout the episode. Each episode lasts for a fixed amount of time. Since the maze layout is randomized at each episode, the agent must learn to quickly remember the maze layout, the target object locations, and the paths leading to them.

**Hyperparameters and Tuning Strategy:** We include the details of the Memory Maze experiments. All of the experiments in that section were implemented using asynchronous PPO implementation from Sample Factory library (Petrenko et al. (2020)). We started with the default hyperparameters for the DMLab lab experiments in Schulman et al. (2015), and finetuned the learning rate and entropy coefficient. For each of LSTM, GTrXL and AGaLiTe, to tune the learning rate and entropy coefficient, we run a sweep for three seeds for 15M steps in the Memory Maze $11 \times 11$ environment. We average the results for the last 1M steps across the three seeds and select the best hyperparameter according to total episodic reward. Using the best-identified hyperparameter, we generate the final results for 100M steps for each of the three seeds. We detail the hyperparameters along with the sweeps for the learning rate and entropy coefficient in Table 7. We include the architecture configuration for each of the 3 architectures in Table 8.

Table 7: Hyperparameters and sweeps for Memory Maze experiments.

| Hyperparameter | Value |
|---|---|
| Learning Rate | [0.0025, 0.00025, 0.000025] |
| Discount Factor ($\gamma$) | 0.99 |
| Advantage Estimation Coefficient ($\lambda$) | 0.95 |
| Entropy Coefficient | [0.03, 0.003, 0.0003] |
| Number of Epochs | 1 |
| Rollout Length | 200 |
| Sequence Length | 100 |
| Batch Size | 3200 |
| PPO Clip Ratio | 0.1 |
| PPO Clip Value | 1 |
| Max Gradient Norm | 4 |
| Value Function Coefficient | 0.5 |
| Number of Workers | 32 |
| Number of Envs per Worker | 2 |

Table 8: Architecture configuration for GTrXL and AGaLiTe for Memory Maze experiments.

| Hyperparameter | GTrXL | AGaLiTe |
|---|---|---|
| Embedding Dimension ($d$) | 512 | 512 |
| Num Heads | 8 | 8 |
| Head Dim ($d_h$) | 64 | 64 |
| Num Layers ($L$) | 4 | 4 |
| Memory Size ($M$) | 256 | N/A |
| Projection Hyperparameter ($\eta$) | N/A | 4 |
| Approximation Hyperparameter ($r$) | N/A | 7 |

### G.5 Craftax

**Hyperparameters and Tuning Strategy:** The architecture sizes used for Craftax experiments are kept same as in Table 4. We detail the hyperparameters used for the PPO algorithm that used for training the agents in the Craftax environment in Table 9. We tune learning rate and entropy coefficient for the sweeps mentioned in Table 9. Our hyperparameter tuning strategy is as follows: we train 5 seeds per architecture for each the hyperparameter configuration for 100M steps in the Craftax symbolic environment. Finally, we identify the best hyperparameter configuration according to the best episodic reward in the last 1M training steps.

Table 9: Hyperparameters and sweeps for Craftax experiments.

| Hyperparameter | Value |
| --- | --- |
| Learning Rate | [0.0002, 0.0003, 0.0004] |
| Discount Factor ($\gamma$) | 0.999 |
| Advantage Estimation Coefficient ($\lambda$) | 0.8 |
| Entropy Coefficient | [0.0, 0.01, 0.001] |
| Number of Epochs | 4 |
| Rollout Length | 128 |
| Sequence Length | 128 |
| Number of Env | 1024 |
| Batch Size (Sequence Length $\times$ Number of Env) | 130944 |
| Number of Mini Batches | 8 |
| PPO Clip Ratio | 0.2 |
| Max Gradient Norm | 1.0 |
| Value Function Coefficient | 0.5 |

## H   Additional Learning Curves on Smaller Memory Maze Configurations

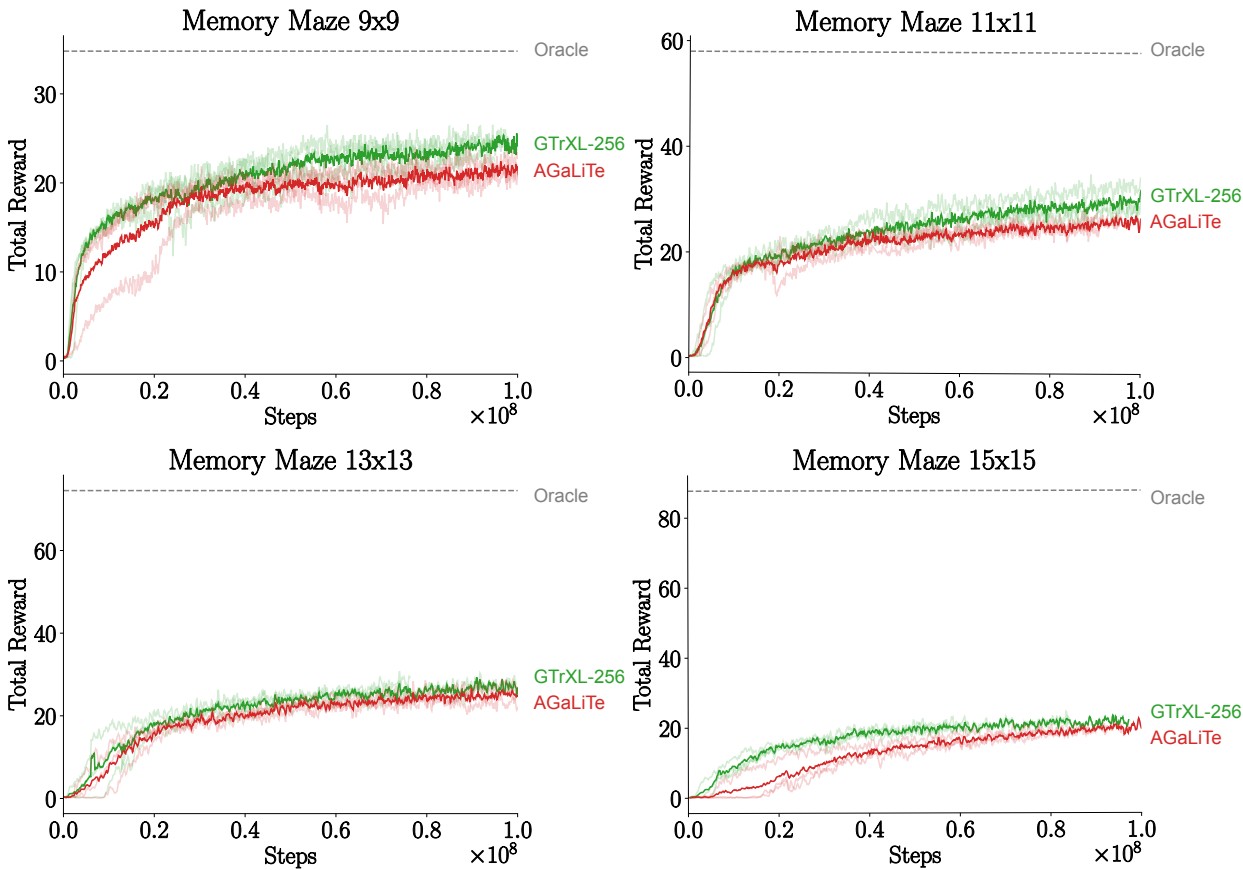

Figure 12: Learning curves of GTrXL and AGaLiTe agents in the Memory Maze environment. The x-axis represents the number of environment steps, and the y-axis represents the total reward in an episode. Each agent is trained with 3 different random seeds. The bold lines represent the mean return across the 3 seeds, and the blurred lines represent the individual seeds. Each point is the average episodic reward over 1M environment steps. The dotted grey line represents the performance of an oracle agent that has access to the entire maze layout, target object locations and paths leading to them.

## I   Evaluating Impact of GTrXL's Context in Memory Maze

This experiment evaluates the impact of GTrXL's context length in the Memory Maze environment. We showed earlier that GTrXL's performance is bottlenecked by the memory size in T-Maze. Our hypothesis is that a similar conclusion should hold in the Memory Maze environment. We expect that GTrXL with a larger memory size would outperform GTrXL with a smaller memory size. We should also be able to show that an AGaLiTe would outperform a GTrXL with a small memory size. To investigate this, we train two additional GTrXL agents with memory sizes of 64 and 128 in the Memory Maze $13 \times 13$ environment.

The learning curves of training the three memory sizes of GTrXL and AGaLiTe in the Memory Maze $13 \times 13$ environment is shown in Figure 13. Asymptotically, all four agents achieve similar performance. The individual learning curves, however, indicate that the GTrXL-64 agent is slower to converge than the GTrXL-128 and GTrXL-256 agents.

The results failed to provide sufficient evidence to support our hypothesis. The performance obtained by the three agents does not appear to be different. This observation leads us to the following speculation:

the Memory Maze environment is too difficult for the agents to be able to utilize their long-term memory capabilities. The reward signal is sparse, which might make it difficult for the agent to learn long-term dependencies. It is also possible that learning long-term dependencies in navigation tasks is harder, in general, and longer training is necessary for the benefits of long-term memory to show.

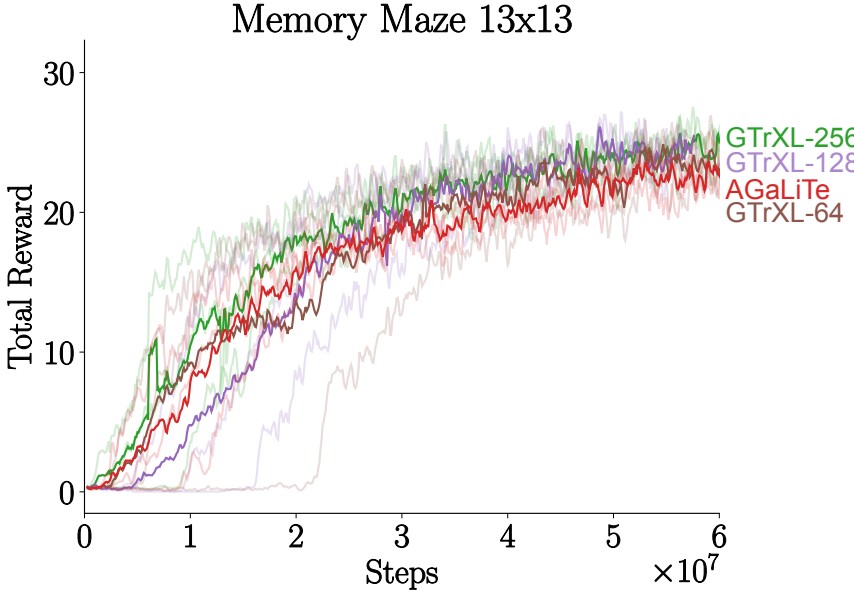

Figure 13: Learning curves of GTrXL agents with different memory sizes in the Memory Maze $13 \times 13$ environment. The x-axis represents the number of environment steps, and the y-axis represents the total reward in an episode. Each agent is trained with 3 different random seeds. The bold lines represent the mean return across the 3 seeds, and the blurred lines represent the individual seeds. Each point is the average episodic reward over 1M environment steps.

## J    Learning curves for various achievements in Craftax Symbolic

In Figure 14, we plot the various achievements in the Craftax environement. Detailed description of these achievements could be found in Hafner (2021).

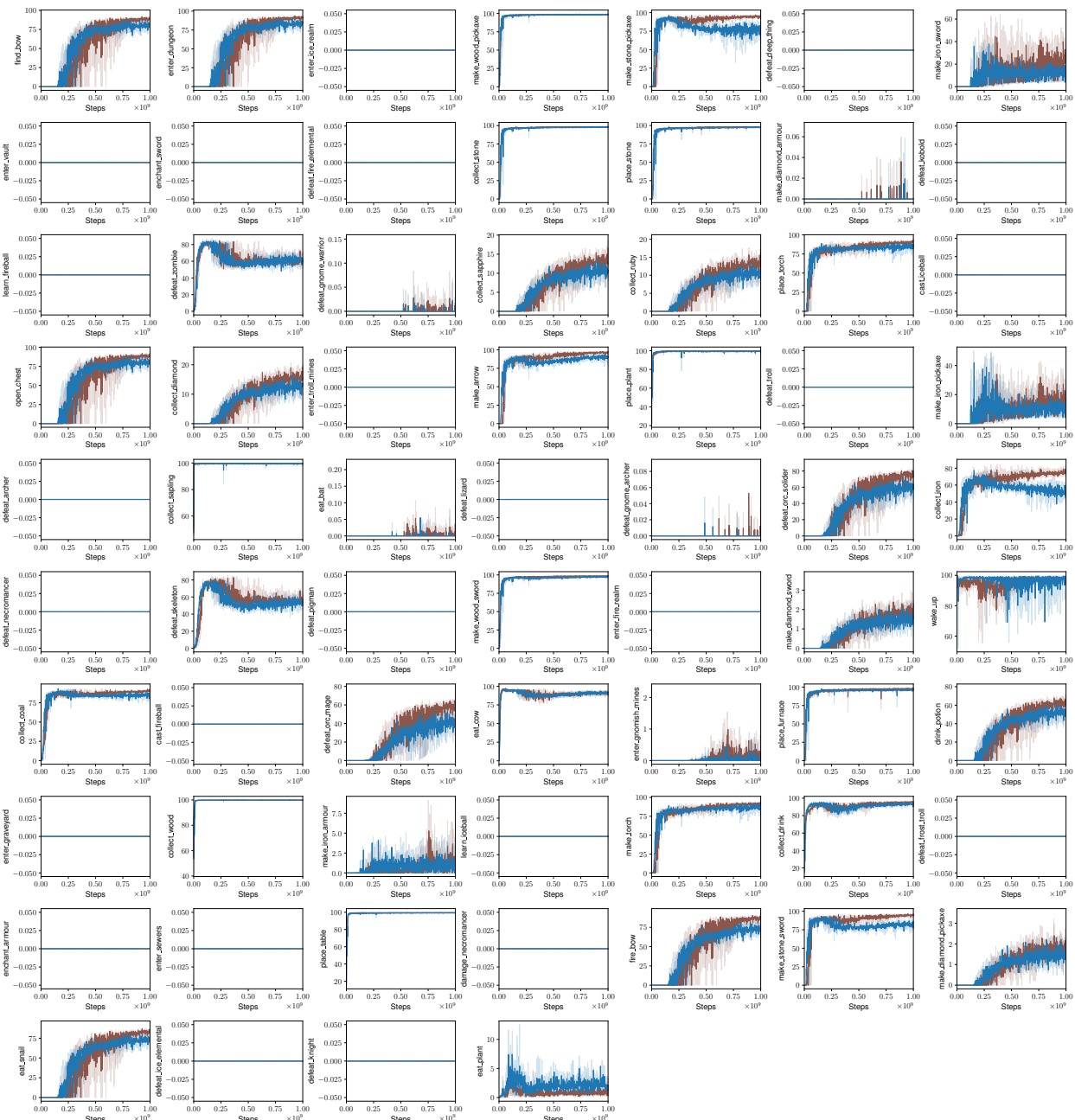

Figure 14: Learning curve for various achievements in Craftax Symbolic. Results are reported over 15 seeds ± 95% bootstrapped CI. AGaLiTe: brown, GTrXL-128: blue.

# K    Latency Measurements

In this section we provide additional empirical evidence of the computational efficiency of our proposed approach, by comparing the latency of forward pass using GTrXL and AGaLiTe. We measure the time required in milliseconds (ms) to do a forward pass in two scenarios: (1) processing single element in streaming sequence, (2) processing an entire sequence in parallel. We configure the architecture sizes of GTrXL and AGaLiTe according to the values used by Parisotto et al. (2020): 12 layers, 8 heads, $d_h = 64$, $d = 256$. We collected all data in a single Google Cloud instance with NVIDIA A100 GPU, 12 CPUs and 80GB RAM.

First, we compare the time required in milliseconds (ms) to do a forward pass using a single element in streaming sequence. We present the results of these comparisons in Figure 15a. According to Dai et al. (2019), XL attention used in the GTrXL architecture has a limited context. The context length of XL attention, how far back in time the transformer architecture can remember, is $\mathcal{O}(ML)$, where $L$ is the number of layers and $M$ is the memory size. We measure the impact of increasing the context length (varying $M$) of GTrXL (x-axis) on the latency to do a single forward pass (y-axis). AGaLiTe does not explicit hyper-parameter that allows controlling the context length, and the use of a recurrent hidden state allows for a potentially unlimited context. Therefore, we consider three AGaLiTe architectures with feature map hyper-parameter $\eta \in [4, 8, 16]$, and plot it as a straight line. We observe that the gap between GTrXL and AGaLiTe increases dramatically with increasing context length.

Next, we measure the time required to do a forward pass over a batch, that is process an entire input sequence in parallel. We present the results of these comparisons in Figure 15b. We vary the length of the input sequence (x-axis) and measure the time required to do a forward pass over the entire sequence (y-axis). We consider two GTrXL architectures with memory size $M \in [128, 512]$. We consider three AGaLiTe architectures with $\eta \in [4, 8, 16]$. We observe that the gap between GTrXL and AGaLiTe increases dramatically with increasing sequence length.

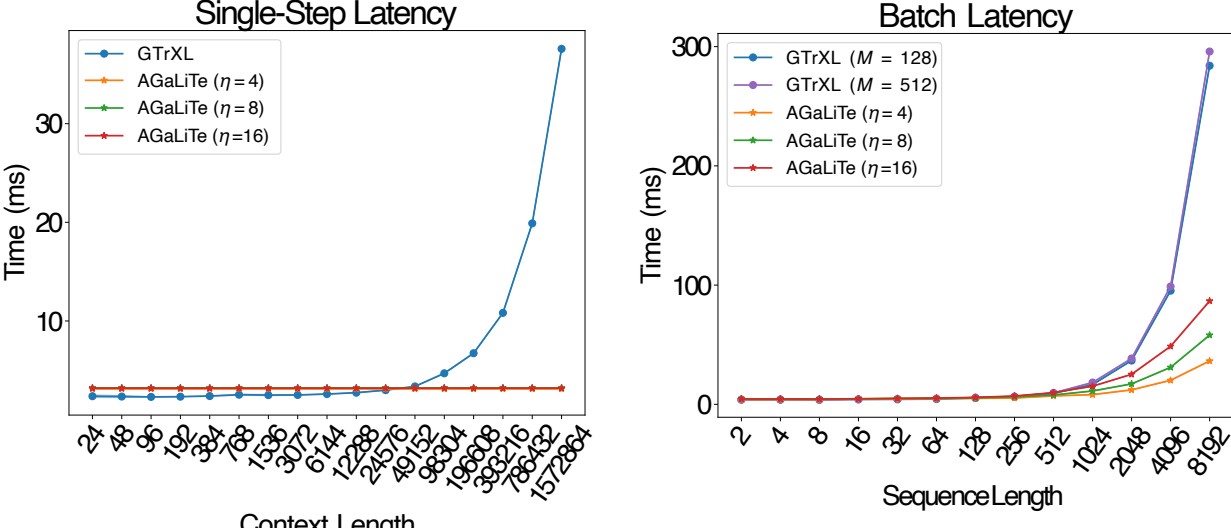

(a) Time (ms) for processing a single element in sequence.

(b) Time (ms) for processing an entire sequence in parallel.

Figure 15: Latency measurements for GTrXL and AGaLiTe. Each point is averaged over 100 independent runs, and the shaded region is the standard error.

