# OpenReview forum: "AGaLiTe: Approximate Gated Linear Transformers for Online Reinforcement Learning"
_TMLR — Accepted by TMLR_

### Review · Reviewer_i1t2 · 2024-08-17

**Summary Of Contributions:**

This paper presents Recurrent Linear Transformer (ReLiT), a new architecture that aims to address the two limitations of linear transformers for reinforcement learning (RL). Specifically, ReLiT introduces a gating mechanism and a learnable feature map. The gating mechanism allows better control of the information flow within the context; the learnable feature map eliminates the choice of the kernel feature map, enhancing the expressiveness of the model. In addition, the authors present an approximate ReLiT as a practical implementation, which reduces the time and space complexity of processing a single element in a sequence. In a variety of environments, ReLiT has shown better performance than other baseline methods.

**Audience:**

Yes

**Claims And Evidence:**

Yes

**Requested Changes:**

Please refer to the section above.

**Strengths And Weaknesses:**

### Strengths
* The writing is generally clear and easy to follow. I have checked most of the formulas in Section 3, and they look correct.

### Weaknesses and other questions
* The performance improvement in the challenging Crafter environment is
* In the paragraph under Section 3, the authors mentioned that the recurrent equations in Algorithm 2 (line 5 and 6) add positive values to the recurrent state. From the equations alone, it seems not necessarily the case. Is it because in linear transformers, the function $\phi$ always takes some special form?
* Section 3.1 "leading to numerical stability issues": is there any evidence or reference to support this statement?
* For the results in Figure 1, could the authors explain why the performance of linear transformers drops significantly at length=200?

---

> ### Author Response · Authors · 2024-08-30
> **Response to Reviewer i1t2**
>
> Thank you for carefully reading our paper and giving thoughtful comments and suggestions. We will respond to the suggested changes and answer your questions below point-by-point.
>
> > The performance improvement in the challenging Crafter environment is
>
> Perhaps there is missing text in this comment. Notice that the results we reported in the submitted version used a smaller architecture size and different hyperparameters than those reported by Matthews et al. (2024). Since then, we have evaluated our approach in this other setting, and we have achieved even better performance. Specifically, we used an architecture of similar size to Matthew et al.'s, and the same discount factor of 0.999, and we trained our model for 1B steps, as done by Matthews et al. (2024). We found that AReLiT achieves a mean episodic reward of 38.89, whereas GTrXL achieves 35.04. Both of these scores are higher than the previously reported PPO+RNN baseline which achieves 34.57 [Matthews et al. (2024)]. We have now updated the paper with these new results.
>
> > In the paragraph under Section 3, the authors mentioned that the recurrent equations in Algorithm 2 (line 5 and 6) add positive values to the recurrent state. From the equations alone, it seems not necessarily the case. Is it because in linear transformers, the function
> always takes some special form?
>
> Yes, the recurrent equations in Algorithms 5 and 6 indeed add positive values to the recurrent state. We assume that $\phi$ has a positive output, resulting in the $\mathbf{k}_t$ vector having positive values. This results in arbitrary positive values being added to Equation 6 at each iteration. Similarly, if the value vector has positive elements, Equation 5 could also result in positive values being added in each iteration.  We have rewritten this text in the paper to make it more clear (highlighted in blue).
>
> > Section 3.1 "leading to numerical stability issues": is there any evidence or reference to support this statement?
>
> In preliminary experiments, we observed that adding positive values and not having a bound on the value vectors did lead to very large values generated by Equations 5 and 6. Nevertheless, those results were performed in an exploratory phase and we do not report them. The primary motivation for the proposed gating mechanism is that it allows the network to selectively decay past information in the recurrent state. This is useful in tasks that require the agent to selectively filter and update multiple pieces of information in a sequence. We have modified the paper to make this motivation as the primary motivation for the gating mechanism.
>
>
> > For the results in Figure 1, could the authors explain why the performance of linear transformers drops significantly at length=200?
>
> We found this behavior of the performance dropping a particular context length to be typical across all architectures in the T-Maze environment. Similar behavior has been observed in previously reported results in T-Maze (Bakker, 2001).
>
> We hypothesize, that due to the limited effective memory capacity of the original linear transformer (and in other architectures), it becomes difficult for it to recover the cue signal after a certain context length. In the original Linear Transformer architecture, the ELU+1 feature is shown to have a limited memory capacity  (Schlag et al., 2021). We observed that the ELU+1 feature map performs the worst in the ablation conducted in Figure 6 b. In T-Maze, we measure the success rate, which is the rate at which the agent is correctly able to recover the cue signal. As the context length grows, limited memory capacity makes it increasingly difficult for the agent to recover the cue signal from the recurrent state, therefore causing a sudden drop in performance.

---

> > ### Comment · Reviewer_i1t2 · 2024-09-09
> >
> > Thank you to the authors for addressing my questions and comments. My apologies for the incomplete comment earlier. I intended to mention that the performance improvement in Craft seems almost negligible. However, the new experimental results provided in the rebuttal sufficiently address my concern.

---

> > > ### Author Response · Authors · 2024-09-09
> > >
> > > Thank you.

---

### Review · Reviewer_cyG9 · 2024-08-19

**Summary Of Contributions:**

The paper introduces a new transformer architecture called Recurrent Linear Transformers (ReLiT), as well as Approximate Recurrent Linear Transformer (AReLit), to address shortcomings of transformer related to access of past information and expensive inference costs. The paper argues that the two shortcomings inhibit the application of transformers to RL settings, which motivates ReLiT and AReLiT and applies them to solve a number of online RL problems.

The paper first starts by providing an introduction to the application of transformers for RL in Section 1. Section outlines preliminaries of the transformer architecture, including the self-attention mechanism and the Linear Transformer. Section 3 introduces ReLiT with relevant details on how the Linear Transformer architecture is modified with a gating mechanism and learnable feature map. Section 4 describes AReLiT, which replaces the update of the recurrent state using a sum of cosine functions that approximate the Kronecker delta function. Section 5 shows an empirical evaluation on a series of online RL problems that require memory to be solved effectively. ReLiT and AReLiT generally perform fairly well in the studied examples providing competitive performance with better computational efficiency. Section 6 details an ablation of AReLiT showing that importance of different components of the proposed architecture. Section 7 describes related work and Section 8 provides a conlusion.

**Audience:**

Yes

**Broader Impact Concerns:**

The paper could be strengthened with a broader impact section.

**Claims And Evidence:**

No

**Requested Changes:**

* It would be good if the paper can provide additional experiments to mitigate the weaknesses mentioned, as this would strengthen the general validity of the claims for the proposed architecture.
* I think it would be good for the paper to provide a deeper discussion on applying transformers for offline RL settings, as well as pretraining plus finetuning [1] training settings, and motivate how the ReLiT could be useful there. On top of that, it would generally be helpful for the paper to provide a stronger motivation for why solving online RL problems is important since that is the primary focus of the current experiments.

Nice to Have:
* I would some empirical numbers on the training time of ReLiT compared to the baselines in addition to the forward pass study in Appendix K.

[1] Ghugare, Raj, et al. "Searching for High-Value Molecules Using Reinforcement Learning and Transformers." The Twelfth International Conference on Learning Representations.

**Strengths And Weaknesses:**

**Strengths:**
* The paper proposes a practical improvement on transformers that makes them more amendable for online RL problems.
* The details of ReLiT and AReLiT are described in good detail and clarity.
* The paper provides relevant details for the experiments with the ablation study providing evidence for the benefits of the proposed modification.

**Weaknesses:**
* The cases studied for the paper are limited to online only. It would be more convincing if the authors provided evidence that ReLiT and AReLiT can be useful for other ML problems. The paper mentions text classification in Section 5, but does not present any results on it. It would be good to have those results to see if the proposed method applies beyond the studied RL settings.
* The experiments present evidence for problems that the baseline methods can solve. It would be more convincing if the authors showed cases where ReLiT's benefits of memory are critical. One such potential example could be extending T-Maze beyond the memory size of  GTrXL-256.

---

> ### Author Response · Authors · 2024-08-30
> **Response to Reviewer cyG9: 1/2**
>
> Thank you for carefully reading our paper and making thoughtful comments and suggestions. We will respond to the suggested changes and answer your questions below point-by-point.
>
> > The paper mentions text classification in Section 5, but does not present any results on it. It would be good to have those results to see if the proposed method applies beyond the studied RL settings.
>
> We do provide results outside reinforcement learning tasks, in supervised learning tasks. We show our results apply beyond the RL setting. The results for other ML tasks are available in Appendix D. The text-based tasks we considered were two problems from the Long Range Arena benchmark: ListOps and Text (IMDB).
>
> We found that our proposed approach outperforms the Transformer and Linear Transformer in both of these tasks despite using a smaller number of learnable parameters than previously reported baselines. We have now clearly referenced those results in the main text.
>
> > The experiments present evidence for problems that the baseline methods can solve. It would be more convincing if the authors showed cases where ReLiT's benefits of memory are critical. One such potential example could be extending T-Maze beyond the memory size of GTrXL-256.
>
> We already showcased this in Figure 1 using the results of GTrXL-128. We show that due the limiting memory mechanism of GTrXL-128, GTrXL-128 fails when the corridor length exceeds the memory size of the GTrXL architecture (i.e. >128). Further, the computational requirements of GTrXL-128 is much higher than AReLiT. For a single attention head, AReLiT uses roughly $62.67$ times fewer operations and $18.28$ times less space than GTrXL-$128$. We have now noted this detail in the paper.
>
> Finally, we would like highlight that we have obtained new results for Crafter which show higher performance than the GTrXL. The results we reported in the submitted version used a smaller architecture size and different hyperparameters than those reported by Matthews et al. (2024). Since then, we have evaluated our approach in this other setting, and we have achieved even better performance. Specifically, we used an architecture of similar size to Matthew et al.'s, and the same discount factor of 0.999, and we trained our model for 1B steps, as done by Matthews et al. (2024). We found that AReLiT achieves a mean episodic reward of 38.89, whereas GTrXL achieves 35.04. Both of these scores are higher than the previously reported PPO+RNN baseline which achieves 34.57 [Matthews et al. (2024)]. We have now updated the paper with these new results.
>
>
> > It would be good if the paper can provide additional experiments to mitigate the weaknesses mentioned, as this would strengthen the general validity of the claims for the proposed architecture.
>
> As mentioned in the previous comment, the results of applying our proposed approach to other ML problems have been presented in Appendix D. We would like to highlight that the paper aims to evaluate the effectiveness of the proposed architecture only in the context of online RL.

---

> ### Author Response · Authors · 2024-08-30
> **Response to Reviewer cyG9: 2/2**
>
> > I think it would be good for the paper to provide a deeper discussion on applying transformers for offline RL settings, as well as pretraining plus finetuning [1] training settings, and motivate how the ReLiT could be useful there.
>
> We thank the reviewer for this suggestion. We have added a discussion to paragraph 8, section 7, covering some of the existing approaches that apply transformers to other RL paradigms such as offline RL, in-context RL, and pre-training plus finetuning approaches (highlighted in blue).
>
>
> > On top of that, it would generally be helpful for the paper to provide a stronger motivation for why solving online RL problems is important since that is the primary focus of the current experiments.
>
> Online reinforcement learning is well-established problem of study with a broad literature [1][2][3] and is critical many application areas, including game-playing [4][5], robotics control (simulated and real) [6][7] and continual reinforcement learning [8].
>
> [1] Schulman, J., Wolski, F., Dhariwal, P., Radford, A. and Klimov, O., 2017. Proximal policy optimization algorithms. arXiv.
>
> [2] Mnih, V., Kavukcuoglu, K., Silver, D., Rusu, A.A., Veness, J., Bellemare, M.G., Graves, A., Riedmiller, M., Fidjeland, A.K., Ostrovski, G. and Petersen, S., 2015. Human-level control through deep reinforcement learning. Nature.
>
> [3] Sutton, R. S., & Barto, A. G., 2018. Reinforcement learning: An introduction (2nd ed.). MIT Press
>
> [4] Mnih, V., Kavukcuoglu, K., Silver, D., Rusu, A.A., Veness, J., Bellemare, M.G., Graves, A., Riedmiller, M., Fidjeland, A.K., Ostrovski, G. and Petersen, S., 2015. Human-level control through deep reinforcement learning. Nature.
>
> [5] Vinyals, O., Babuschkin, I., Czarnecki, W.M., Mathieu, M., Dudzik, A., Chung, J., Choi, D.H., Powell, R., Ewalds, T., Georgiev, P. and Oh, J., 2019. Grandmaster level in StarCraft II using multi-agent reinforcement learning. Nature.
>
> [6] Haarnoja, T., Zhou, A., Hartikainen, K., Tucker, G., Ha, S., Tan, J., Kumar, V., Zhu, H., Gupta, A., Abbeel, P. and Levine, S., 2018. Soft actor-critic algorithms and applications. arXiv
>
> [7] Kober, J., Bagnell, J.A. and Peters, J., 2013. Reinforcement learning in robotics: A survey. The International Journal of Robotics Research.
>
> [8] Khetarpal, K., Riemer, M., Rish, I. and Precup, D., 2022. Towards continual reinforcement learning: A review and perspectives. Journal of Artificial Intelligence Research.
>
>
>
> > I would some empirical numbers on the training time of ReLiT compared to the baselines in addition to the forward pass study in Appendix K.
>
> Yes, we have already done this in Memory Maze results in Section 5. AReLiT achieves $535.63 \pm 0.52$ FPS while GTrXL achieves $373.63 \pm 0.49$ FPS, corresponding to a 43.36% improvement in training time over GTrXL.

---

> > ### Comment · Reviewer_cyG9 · 2024-09-15
> >
> > Thank you for the additional details and revisions. Most of my feedback has been addressed.

---

> > > ### Author Response · Authors · 2024-09-16
> > >
> > > Thank You.

---

### Review · Reviewer_E4UB · 2024-08-22

**Summary Of Contributions:**

The paper argues that the existing linear transformer architectures have important limitations – (1) the self attention mechanisms adds positive values to the recurrent state and therefore cannot delete previous information; (2) performance critically depends on the choice of the kernel function; (3) kernel functions often use element-wise or randomized expansive feature maps, which can result in limited memory capacity; (4) they have a high memory cost due to the maintenance of a matrix as a recurrent state.

The paper proposes a novel architecture (learning framework) that addresses these limitations by (i) introducing a gated structure that can focus on relationships with a much larger time gap; (ii) incorporating a learnable feature map in the self-attention mechanism (ensuring that it is amenable to sequential computation); and (iii) proposing an approximate version that eliminates the need to maintain a matrix as the recurrent state.

The paper provides an extensive set of experiments focusing on the partially-observable reinforcement learning task to demonstrate a performance improvement compared to the state-of-the-art transformer-based RL method.

In the appendix, the paper provides a proof that the limit of the approximating function is the Kronecker delta, as well as thorough details of the experiments and additional experimental results.

**Audience:**

Yes

**Broader Impact Concerns:**

No concerns

**Claims And Evidence:**

Yes

**Requested Changes:**

(1)	Although I believe it is important, weakness 1 is easily rectified by clearer identification in the introduction that the paper is focused on the RL setting and the claims apply to that setting. I don’t consider that this significantly diminishes the contribution of the paper. The alternative is to report the results of experiments addressing other tasks (in particular, tasks similar to those that have been explored in some of the papers that introduces the baseline techniques). Reinforcement learning is not explicitly mentioned in the first four paragraphs of the introduction. Yes, the first paragraph describes an RL setting, but the next paragraph switches to the more general task of “incremental state-construction”. The paper needs to explicitly specify that the identified weaknesses and rectifications are in the context of the RL setting (no experimental evidence is provided that the proposed techniques are beneficial for other tasks, so the claim is not supported in the general setting). For this reason, I would also suggest that the title is modified to include “Reinforcement Learning”, but this is only a suggestion. Mandatory: modify the text to more clearly specify the context of the claims.

(2)	I do not completely understand the choice of the name of the architecture. The linear transformer is already recurrent (as the paper clear specifies in Section 2.2 – “maintains a matrix … as a recurrent state”). How is a “recurrent linear transformer” different from a linear transformer? “Recurrent” is an unnecessary redundancy that doesn’t differentiate the proposed architecture in a clear way. Yes, the paper proposes a modification to the recurrence equations, but it doesn’t introduce recurrence, which is what the name implies. Beyond this, there are several works (discussed below) that modify the recurrence in other ways, or aim to introduce stronger recurrence relationships. Please either explain the choice of the name or consider a different, more informative and appropriate name.

(3)	The discussion of the related work should be expanded and the connections and distinctions of the proposal compared to previously introduced methods made much clearer.

The paper cites Schlag et al. 2021, primarily in support of its motivations, but it does not provide a clear and thorough discussion of the relationship of the proposed method to the methods in that work. Nor does it cite or discuss the follow-up work [R1]. These two works introduce the DeltaNet, which does address the memory capacity concern, by allowing for both positive and negative updates and incorporating a learnable learning rate (although not element-wise). The related work in the submitted paper does explain that the key difference is that the proposed method includes an element-wise gate, although it is not acknowledged clearly that other techniques already address the positive addition and memory limitation.

In [R1], several extensions are presented that incorporate additional recurrence. The Delta RNN applies “an element-wise activation function f to the previous output of the fast network y(t−1) to obtain the recurrent query” – there is a separate set of weights for the recurrence. A Delta LSTM is also proposed. The Recurrent Delta Net introduces recurrence into the k, v, q update equations, as well as the scalar gate.

The paper should also cite and discuss [R2]. This is probably concurrent, independent work (the paper was arXived in Feb. 2024, but only published at an August 2024 conference). Now that it has been published, the paper should certainly be discussed in the related work section, probably with a statement that the research was conducted concurrently and independently. As the title implies, [R2] proposes the use of learnable kernels (“enabling the model to learn any quadratic function that is non-negative and has a single real root”).

[R1] Irie, Kazuki, et al. "Going beyond linear transformers with recurrent fast weight programmers." Advances in Neural Information Processing Systems 34 (2021): 7703-7717.

[R2] Yaroslav Aksenov, Nikita Balagansky, Sofia Lo Cicero Vaina, Boris Shaposhnikov, Alexey Gorbatovski, and Daniil Gavrilov. 2024. Linear Transformers with Learnable Kernel Functions are Better In-Context Models. In Proceedings of the 62nd Annual Meeting of the Association for Computational Linguistics (Volume 1: Long Papers), pages 9584–9597, Bangkok, Thailand.

Questions

(Q1)	In Figure 2, the AReLiT behaviour seems considerably less stable than the GTrXL. Is this correct observation? If so, is there an explanation? The GTrXL also seems to still be improving at the selected endpoint. Does it reach/surpass the behaviour of AReLiT after more steps?

(Q2)	Is there a reason that a 2020 paper (Parisotto et al., 2020) is still the state-of-the-art baseline? Is there a need to incorporate other baselines that are not based on transformers?

(Q3) “Note that we present the algorithm for processing a sequence in the case of the linear transformer. This is in contrast to Algorithm 1, which presents the algorithm for processing a sequence.” – I suspect this is a typo and it is the processing of a single input vector?

**Strengths And Weaknesses:**

Strengths:

S1.	The paper introduces three novel architectural innovations.

S2.	The paper is clearly written, with an excellent motivation for the introduced techniques, and a detailed exposition of the methodology.

S3.	Experiments are well-formulated and provide compelling evidence in support of the claims of the paper in the context of reinforcement learning tasks.

S4.	The approximation to the Kronecker delta function is innovative and well-motivated, the paper provides a proof of convergence, and the resultant architecture with a reduced memory footprint performs well.

Weaknesses:

W1.	To substantiate the claims, the paper needs to establish that the specified attributes of the existing architectures are indeed limitations and that they are overcome or mitigated by the proposed architecture. The paper successfully does this for partially observable RL problems, but it does not provide any results for other tasks. For example, Katharopoulos et al. (2020) report on image generation and speech recognition experiments.

W2.	Insufficient explanation for the name of the architecture and potential confusion with both the original linear transformer and other more recent works that adapt the recurrence relationships in other ways.

W3.	The paper does not sufficiently discuss the relationship to important related work that presents techniques similar to some of the introduced methods.

---

> ### Author Response · Authors · 2024-08-30
> **Response to Reviewer E4UB: 1/2**
>
> Thank you for carefully reading our paper and giving thoughtful comments and suggestions. We will respond to the suggested changes and answer your questions below in a point-by-point fashion.
>
> > (1) Although I believe it is important, … of “incremental state-construction … context of the claims.”.
>
> We agree with the reviewer that the experiments presented in this paper demonstrate the effectiveness of our proposed approach mainly in the context of partially observable RL. We have modified the paper's introduction to make the paper's focus and problem setting more explicit. Additionally, we have changed the paper's title to: “AGaLiTe: Approximate Gated Linear Transformers for Online Reinforcement Learning”, to state more clearly that the proposed approach is in the context of reinforcement learning tasks. We describe the decision to change the name of the architecture in our next comment.
>
> > (2) I do not completely understand the choice of the name of the architecture. … Please either explain the choice of the name or consider a different, more informative and appropriate name.
>
> We agree with the reviewer that the reviewer that the proposed approach does not introduce recurrence to linear transformers. During our initial experiments, we found that our proposed gating mechanism specifically helps in the recurrent training of linear transformers using truncated backpropagation, and therefore the word “recurrent” stayed. Nevertheless, those results were performed in an exploratory phase.
>
> We have modified the name of the architecture to Gated Linear Transformer (GaLiTe) and Approximate Gated Linear Transformer (AGaLiTe) to more accurately reflect the proposed mechanisms in our architecture.
>
> >(3) The discussion of the related work should be expanded and the connections and distinctions of the proposal compared to previously introduced methods made much clearer.
>
>
> We thank the reviewer for suggesting additional related work. We have now included and discussed the related works: Schlag et al. 2021, [R1], and [R2]. We have added this discussion in paragraph 1, section 7 of the updated paper (highlighted in blue).

---

> ### Author Response · Authors · 2024-08-30
> **Response to Reviewer E4UB: 2/2**
>
> > (Q1) In Figure 2, the AReLiT behaviour seems considerably less stable than the GTrXL. Is this correct observation? If so, is there an explanation? The GTrXL also seems to still be improving at the selected endpoint. Does it reach/surpass the behaviour of AReLiT after more steps?
>
> We tried to dig deeper to see if this observation was accurate. So, we looked at the individual seeds for both and didn’t notice differences in seed variability between GTrXL and AReLiT.
>
> We also ran the experiments for longer (10 million steps), and even though GTrXL performance improved, AReLiT was still outperforming GTrXL. GTrXL’s performance indeed plateaus after 5M steps, as we previously reported, but when we run it for longer we see that it plateaus in a lower value than AReLiT’s.
>
> Link to results: https://docs.google.com/document/d/1wQnvmevNJ1Ex8xy-yQWTtsUxGEgPAcfv__dTUJgxTNU/edit?usp=sharing
>
>
>
> > (Q2) Is there a reason that a 2020 paper (Parisotto et al., 2020) is still the state-of-the-art baseline? Is there a need to incorporate other baselines that are not based on transformers?
>
> In the partially observable reinforcement learning setting, the work by Parisotto et al. (2020) is still the most recent baseline that proposes architectural changes to the transformer architecture to make it amenable to the online reinforcement learning setting—there is relatively little work on extending transformers to online RL. Perhaps this might be because the transformer architecture is commonly used in tasks where the input sequence is readily available, and the training process can be massively parallelized using GPUs. In the case of RL, the input sequence must be collected through continual interaction with the environment, and parallelization isn’t possible. This makes the application of vanilla transformers in RL less common.
>
> Given the focus of our work on enhancing transformer architectures for online RL, we have primarily considered transformer-based baselines. We would like to highlight that GTrXL is still reported as the best-performing baseline in some of the recent environments considered in this paper: Memory Gym environment (https://arxiv.org/abs/2309.17207) and Craftax (https://github.com/MichaelTMatthews/Craftax) which were published in 2023 and 2024, respectively.
>
> > (Q3) “Note that we present the algorithm for processing a sequence in the case of the linear transformer. This is in contrast to Algorithm 1, which presents the algorithm for processing a sequence.” – I suspect this is a typo and it is the processing of a single input vector?
>
> Correct, this is indeed a typo, and the correct statement should be: “Note that we present the algorithm for processing a single input vector in the case of the linear transformer. This is in contrast to Algorithm 1, which presents the algorithm for processing a sequence.”
> We have modified the paper accordingly.

---

### Decision · Action_Editor_xJeT · 2024-09-27

**Recommendation:** Accept with minor revision

**Comment:**

The authors should be sure to provide a final, camera ready version that includes all of the modifications they have made in revision.

**Audience:**

Yes, the ML field will find the novel architecture for online reinforcement learning in partially observable contexts useful and interesting.

**Claims And Evidence:**

This paper examines architectures for online reinforcement learning in partially observable contexts. More specifically, it proposes a novel recurrent, gated framework that addresses identified limitations of existing linear transformer architectures. The authors claim that this framework offers context-independent inference cost, can leverage long-range dependencies effectively, and performs well in online reinforcement learning tasks. The paper provides an experiments to demonstrate a performance improvement compared to the state-of-the-art transformer-based RL method in a variety of partially observable tasks, such as various maze tasks. They also provide evidence that inference with their approach is more efficient than the current state-of-the-art transformer architecture.

The reviewers were generally positive, though they raised some concerns about the evidence backing the claims, such as concerns around which attributes of the new approach are responsible for the improvements on the tested tasks. There were also some questions related to clarity. After rebuttal, the reviewers were largely satisfied that the evidence supported the claims, and all recommended acceptance.